# Ultrafast On-Chip Online Learning via Spline Locality in Kolmogorov–Arnold Networks

Duc Hoang [* 1]   Aarush Gupta [* 1]   Philip Harris [1]

## Abstract

Ultrafast online learning is essential for high-frequency systems, such as controls for quantum computing and nuclear fusion, where adaptation must occur on sub-microsecond timescales. Meeting these requirements demands low-latency, fixed-precision computation under strict memory constraints, a regime in which conventional Multi-Layer Perceptrons (MLPs) are both inefficient and numerically unstable. We identify key properties of Kolmogorov-Arnold Networks (KANs) that align with these constraints. Specifically, we show that: (i) KAN updates exploiting B-spline locality are sparse, enabling superior on-chip resource scaling, and (ii) KANs are inherently robust to fixed-point quantization. By implementing fixed-point online training on Field-Programmable Gate Arrays (FPGAs), a representative platform for on-chip computation, we demonstrate that KAN-based online learners are significantly more efficient and expressive than MLPs across a range of low-latency and resource-constrained tasks. To our knowledge, this work is the first to demonstrate model-free online learning at sub-microsecond latencies.

## 1. Introduction

Ultrafast **model-free online learning**, where the network learns an input-output mapping directly from streaming data without assuming an explicit system model, is essential in domains with high-frequency, non-stationary dynamics. These include quantum control, high-speed communications, and plasma diagnostics, where adaptation must occur on sub-microsecond timescales (Berritta et al., 2024). In these regimes, host–accelerator training loops are too slow:

*Equal contribution  [1]MIT. Correspondence to: Duc Hoang <dhoang@mit.edu>.

*Proceedings of the $43^{rd}$ International Conference on Machine Learning*, Seoul, South Korea. PMLR 306, 2026. Copyright 2026 by the author(s).

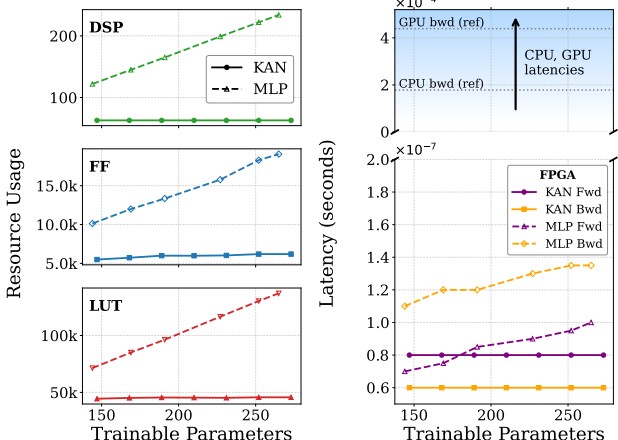

*Figure 1.* **Hardware scaling under ultrafast on-chip online learning.** On a non-stationary qubit readout task with fixed-point training, MLPs grow roughly linearly in on-chip resources (DSP/FF/LUT) and in forward/backward latency with parameter count. KANs leverage B-spline locality and favorable approximation scaling, increasing capacity (via grid size $G$) with near-constant resources and sub-100 ns latency. CPU/GPU (A100) lines show reference latencies for PyTorch implementations.

by the time gradients are computed off-chip and parameters return over high-latency links (e.g. PCIe), the system may have already shifted operating conditions. Real-time adaptation, therefore, *requires* both inference and training to be performed directly on-chip, with low-latency, fixed-precision computation under strict memory constraints.

Currently, end-to-end on-chip training with standard MLPs remains impractical. Backpropagation is expensive, and reduced precision can destabilize gradient-based optimization (Yan et al., 2024). As a result, prior systems focused on either hyperspecialized training workflows (Tang et al., 2022) or a decoupling of learning from inference, even when continuous adaptation is required (Sivak et al., 2025). This motivates a fundamental architectural question: *what model properties make fixed-point, low-latency, on-chip online updates both stable and hardware-efficient?*

We revisit **Kolmogorov–Arnold Networks (KANs) (Liu et al., 2025) from a hardware-centric perspective**, where their perceived inefficiency (Tran et al., 2024; Hou et al.,

2025) largely reflects assumptions about batched, floating-point training and inference on GPUs. In contrast, custom hardware can exploit KANs' locally supported activations and sparse gradient updates, mitigating the global error propagation and dense compute overhead of MLPs.

Guided by theory on per-sample update cost and approximation error, we realize an efficient and ultrafast implementation of KAN on custom hardware (FPGA). Under equal parameter budgets and identical precision constraints, KANs achieve superior numerical stability, lower update cost, and ultrafast latency. This enables reliable gradient updates on streaming data, supporting model-free online learning at sub-microsecond timescales for the first time.

## 2. Related Works

**Application Domains.** Ultra low-latency online adaptation is critical in domains where system dynamics evolve on microsecond or nanosecond timescales, including plasma control (Wei et al., 2024), optical and wireless communications (Wang et al., 2018), quantum control (Reuer et al., 2023; Guglielmo et al., 2025; Liyanage et al., 2024), and particle accelerator tuning (Scheinker, 2021). Most existing approaches rely on meta-learning (Nagabandi et al., 2019) or heterogeneous host–device pipelines, in which learning occurs off-device and models are periodically redeployed (Sivak et al., 2025). Such workflows incur communication and scheduling latency and cannot support deterministic, sub-microsecond adaptation. As a result, while custom hardware is widely used for real-time control (Berritta et al., 2024) and inference (Guglielmo et al., 2025), learning is typically absent or limited to primitive update rules (Jones et al., 2025).

**On-Device Training.** On-device neural network inference is well established across custom and embedded hardware platforms (e.g. HLS4ML (Fahim et al., 2021), FINN (Blott et al., 2018)), but fully on-device training remains largely unexplored. Prior work focuses on accelerating batch training (Dey et al., 2018; Luo et al., 2019) or targets narrow, task-specific scenarios (Liu et al., 2023; Gadea et al., 2000), rather than continuous online adaptation. Backpropagation fundamentally limits this regime by increasing arithmetic workload, requiring dense activation storage, and amplifying numerical instability under reduced precision (Tang et al., 2022; Yan et al., 2024). Existing architectural and compiler-level optimizations mitigate only specific bottlenecks (Zhang et al., 2021; Venkataramanaiah et al., 2019; Kara et al., 2017). Consequently, despite the dominance of MLPs driven by the "hardware lottery" (Hooker, 2020), current approaches remain unsuitable for stable, deterministic on-chip online learning under tight latency and precision constraints.

**KANs and Perceived Hardware Limitations.** KANs (Liu et al., 2025) use learnable spline activations and perform well empirically, yet are often labeled hardware-impractical due to recursive spline evaluation and increased per-edge parameterization (Tran et al., 2024; Hou et al., 2025). However, these prevailing critiques of KAN hardware inefficiency are GPU-centric or inference-focused and overlook the locality and sparsity induced by B-spline activations. Prior hardware work targets analog hardware (Duarte et al., 2025) or LUT-based inference (Hoang et al., 2026; Huang et al., 2025), but does not address training. To our knowledge, fully on-chip KAN training with deterministic execution, fixed-point arithmetic, and bounded local memory has not been analyzed or demonstrated, leaving the question of whether KANs can support hardware-efficient on-device learning entirely open.

**Other Adaptive Methods.** Online adaptation includes statistical estimators and specialized learners (e.g. Kalman/LMS filters, ELMs, RBF, sparse-coding models, and SNNs) that can run at low latency on hardware (Babu & Parthasarathy, 2022; Johnson et al., 2024; Zang et al., 2025; Le et al., 2017; Borrageiro et al., 2022; Mehrabi et al., 2024). However, these methods either assume restricted model classes (e.g. linear/Gaussian), rely on fixed/random feature maps (e.g. ELM/RBF), or require non-standard training pipelines with non-deterministic timing (e.g. SNNs). These properties make them unsuitable baselines for *model-free, gradient-based* online learning with *deterministic* sub-$\mu$s updates. Moreover, prior studies focus on inference or task-specific update rules rather than end-to-end training pipelines.

## 3. Theoretical Motivation

**Setting.** We compare MLPs and KANs under fixed-point quantization, where quantized values are spaced evenly along a static range.

**Notation.** An MLP layer $\ell$ is $\Phi_\ell(x) = \sigma(W_\ell x + b_\ell)$ with $W_\ell \in \mathbb{R}^{d_{out} \times d_{in}}, b_\ell \in \mathbb{R}^{d_{out}}$. A KAN layer $\ell$ applies learnable univariate spline maps on edges:

$$x_{\ell+1,q} = \sum_{p=1}^{n_\ell} \phi_{\ell,q,p}(x_{\ell,p}), \quad \phi_{\ell,q,p}(x) = \sum_{i=1}^{G+S} w_{\ell,q,p,i}\, B_i(x),$$

where $\{B_i\}_{i=1}^{G}$ are $S$-th order B-spline basis functions on a grid of size $G$.

Throughout this analysis, we omit bias terms in MLPs and base activation terms in KANs for notational simplicity. These omissions do not affect the complexity or boundedness results, as both terms follow the same structural arguments as their respective main terms.

### 3.1. B-Spline Locality Enables Sparse Updates

The computational advantage of KANs for on-chip learning stems from B-spline basis functions having local support.

**Lemma 3.1** (Local support of B-splines (Liu et al., 2025)). *For spline order $S$, each basis function $B_i$ has local support. For any input coordinate $x_p$, exactly $S + 1$ indices $i$ satisfy $B_i(x_p) \neq 0$.*

This locality directly reduces per-sample update cost:

**Proposition 3.2** (Update complexity reduction). *Let $\mathcal{C}_{\mathrm{update}}(\cdot)$ denote the number of scalar arithmetic operations required to compute parameter gradients for one sample. Under an equal parameter budget $N$,*

$$\mathcal{C}_{\mathrm{update}}(\mathrm{KAN}) = \frac{S + 1}{G + S} \, \mathcal{C}_{\mathrm{update}}(\mathrm{MLP}).$$

The proof appears in Appendix A.1. The key insight is that while MLPs produce dense gradients over all $N$ parameters, KANs update only the $S + 1$ active coefficients per edge, **independent of total capacity** $N$.

### 3.2. Capacity Scaling at Constant Compute

Proposition 3.2 has a powerful implication for capacity scaling. KAN approximation quality improves with grid resolution: for sufficiently smooth targets and $0 \leq m \leq S$, there exist spline maps on a grid of size $G$ such that (Liu et al., 2025)

$$\left\| f - f_G \right\|_{C^m} \leq C \, G^{-S-1+m}.$$

Crucially, increasing $G$ adds *stored* coefficients but not *active* computation. By Lemma 3.1, each edge still reads and updates only $S + 1$ coefficients per sample. In contrast, scaling MLP capacity requires proportionally more MACs and gradient updates.

This decoupling also benefits continual learning: because each sample modifies only coefficients in a local region of the input space, updates preserve prior fits elsewhere, reducing catastrophic forgetting compared to dense MLP gradients.

**Hardware Takeaway.** On custom hardware, per-sample latency and memory bandwidth scale with the size of the *active* coefficient set $S + 1$, not total parameters $N$. KANs thus expose a clean capacity knob: scaling the grid size $G$ improves approximation capabilities while per-sample compute remains essentially constant. The main cost is extra coefficient storage (memory depth), a favorable trade-off on-chip (LUT/FF/BRAM) that typically adds little overhead compared to arithmetic compute on custom hardware.

### 3.3. Robustness to Fixed-Point Quantization

KANs possess properties that make them more robust to fixed-point quantization noise as compared to MLPs.

**Proposition 3.3** (KAN Activation Bounds). *Consider a KAN activation $\phi(x) = \sum_i W_i B_i(x)$. Then, for any input $x$,*

$$\min_i W_i \leq \phi(x) \leq \max_i W_i.$$

This proposition is proven in Appendix A.2.

**Remark.** Unlike MLPs, where outputs scale with inputs via multiplication, KAN activations are convex combinations of learned coefficients, which are bounded and stable regardless of input magnitude. For a given input range, B-spline functions also tend to produce outputs that are uniformly distributed in magnitude. This inherent *magnitude normalization* in KANs is essential for stability under fixed-point quantization, where representing wide dynamic ranges is challenging.

**Proposition 3.4** (Bounded Gradient Sensitivity). *Consider layer $\ell$, where $g_{\ell+1,q} = \frac{\partial \mathcal{L}}{\partial x_{\ell+1,q}}$ denotes the backpropagated gradient received by that layer.*

**MLP.** *The weight gradient satisfies*

$$\frac{\partial \mathcal{L}}{\partial W_{\ell,q,p}} = g_{\ell+1,q} \cdot x_{\ell,p},$$

*scaling linearly with input magnitude $|x_{\ell,p}|$.*

**KAN.** *The coefficient gradient satisfies*

$$\frac{\partial \mathcal{L}}{\partial w_{\ell,q,p,i}} = g_{\ell+1,q} \cdot B_i(x_{\ell,p}),$$

*where $B_i(x) \in [0, 1]$ by B-spline nonnegativity and partition of unity.*

The proof for this proposition is provided in Appendix A.3.

**Remark.** While Proposition 3.3 bounds *forward* activations within the coefficient range, this proposition bounds *backward* error propagation in gradients within the B-spline envelope. Together, they ensure that both forward and backward passes remain numerically stable under aggressive fixed-point quantization, a critical property for resource-constrained on-chip implementations where wide dynamic range is impractical.

## 4. Custom Hardware Implementation

Figure 2 contrasts our custom FPGA training kernels for KAN and MLP under identical streaming I/O, fixed-point arithmetic, and fully on-chip state. Here we highlight only

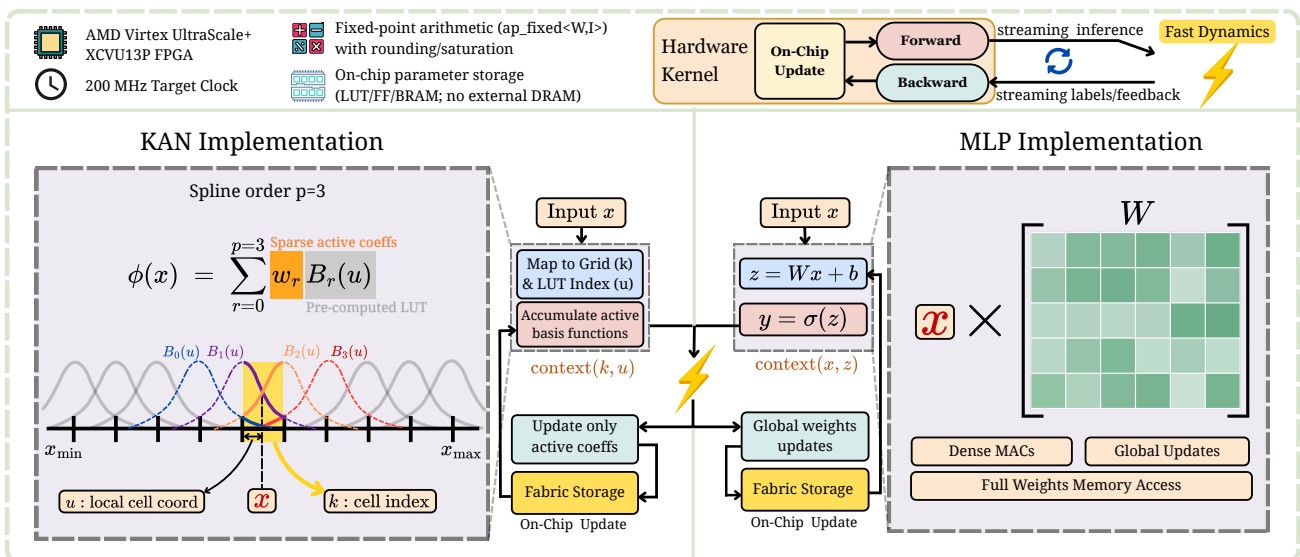

*Figure 2.* **Overview of our streaming custom hardware kernels for fully online learning.** KAN (left) and MLP (right) are synthesized in Vitis HLS as single hardware kernels (200 MHz, AMD Virtex UltraScale+ XCVU13P) that process streaming inputs and perform forward inference, backward gradient propagation, and in-place on-chip parameter updates with deterministic latency. **KAN:** each input $x$ is mapped to a grid cell index $k$ and LUT index $u$; only the $(S+1)$ active B-spline basis functions are read from a small ROM LUT and accumulated, and the cached $(k, u)$ context is reused in backprop to update only active coefficients. Gradients are computed using B-spline derivatives stored in LUTs. **MLP:** layers compute $z = Wx + b$, $y = \sigma(z)$ with dense MACs and global weight/bias updates using cached pre-activations. LUTs and per-sample context buffers are fully partitioned, while trainable parameters reside entirely on-chip (LUTRAM/BRAM/FF depending on synthesis) with explicit array partitioning to expose parallel accesses (no external DRAM).

the key implementation choices. Full details are provided in Appendix B.

**KAN framework exploiting locality.** Prior KAN implementations are GPU-centric and typically evaluate B-splines via expensive de Boor's recursion, which obscures and underutilizes local support. We instead exploit B-spline locality explicitly: spline basis values/derivatives are stored in small precomputed **ROM LUTs**, and training performs **index-driven updates of only the** $(S+1)$ **active coefficients** per edge. Then, per-sample bandwidth/compute scales with the active set rather than the full parameter tensor.

**MLP baseline.** The MLP uses the same streaming interface and update rule, but performs dense $Wx+b$ MACs and dense (global) weight/bias updates, with a small context buffer caching layer inputs and pre-activations.

## 5. Experiments

We evaluate our FPGA kernels in the target regime of real-time learning: batch-size-one updates, non-stationary streams, and fixed-point arithmetic with deterministic latency. We study three fully online benchmarks: drifting regression (sensor calibration drift), adaptive single-shot qubit readout, and non-stationary Acrobot control. We then analyze scalability in Section 6 on higher-dimensional digit classification with continuous angular drift.

### 5.1. Evaluation: Online learning under fixed-point

**Online protocol.** We adopt a standard online learning protocol (Nagabandi et al., 2019): at each time step $t$, the learner receives $\mathbf{x}_t$, produces $\hat{y}_t$, and performs a single update from the local feedback signal (supervised error or RL reward/advantage). To ensure model non-stationarity, we assume the stream is governed by an unobserved context $\tau_t$ that controls both the data distribution and the optimal decision rule, where it may drift stochastically or adversarially.

**Models and baselines.** We use MLPs as the primary baseline, since they are the dominant general-purpose function approximator and the architecture most directly comparable to KANs. More generally, the MLP baseline serves as a proxy for the broader class of non-local architectures (e.g. dense matrix multiplication layers). In each experiment, we compare KANs against two fixed-role MLP baselines: (i) **MLP-P**, parameter-matched to the KAN budget $N$, and (ii) **MLP-L**, a larger MLP scaled to match the KAN's stable asymptotic performance. For Acrobot, we specify the online algorithm, Actor-Critic (Konda & Tsitsiklis, 1999) or DQN (Mnih et al., 2013), when relevant. For MLPs, we sweep depth/width and ReLU, Tanh, and SiLU activations, and report the best-performing design under the same online and numeric constraints.

We focus on streamlined MLPs because auxiliary techniques such as LayerNorm impose substantial hardware costs under

*Table 1.* **Experimental configurations.** All models use single-batch updates. Fixed-point formats are reported as $\langle W, I \rangle$. Learning rates and architectures are grid-searched; ReLU outperforms Tanh/SiLU on hardware. Metric reports the task objective: cumulative regret ($\downarrow$) for regression, accuracy ($\uparrow$) for classification, and episodic return ($\uparrow$) for control.

| Experiment | Model | Dimension | Activation | # Params | $G$ | $s$ | $\eta$ | Bitwidth | Metric ($\downarrow$**Reg** / $\uparrow$**Acc** / $\uparrow$**Ret**) | Task |
|---|---|---|---|---|---|---|---|---|---|---|
| Adaptive Function Approximation | KAN | [1,1] | - | 13 | 10 | 3 | 0.5 | $\langle 6, 2 \rangle$ | **13.2** | Regression |
| | MLP-P | [1,2,2,1] | ReLU | 13 | – | – | 0.1 | $\langle 6, 2 \rangle$ | 97.6 | Regression |
| | MLP-L | [1,16,16,1] | ReLU | 321 | – | – | 0.1 | $\langle 6, 2 \rangle$ | 48.3 | Regression |
| Single-Shot Qubit Readout | KAN | [2, 7, 1] | - | 273 | 10 | 3 | 0.05 | $\langle 7, 3 \rangle$ | **92.8%** | Classification |
| | MLP-P | [2,20,8,5,1] | ReLU | 279 | – | – | 0.01 | $\langle 7, 3 \rangle$ | 50.4% | Classification |
| | MLP-L | [2,16,16,1] | ReLU | 609 | – | – | 0.015 | $\langle 7, 3 \rangle$ | 49.0% | Classification |
| | OS-ELM | 10 hidden features | Sigmoid | – | – | – | – | float | 41.8% | Classification |
| Non-Stationary Acrobot Control | KAN-AC | [6,4] | - | 144 | 5 | 1 | 1e$-$3 | $\langle 22, 8 \rangle$ | **-104** | Control (RL) |
| | MLP-AC-P | [6,13,4] | ReLU | 147 | – | – | 3e$-$4 | $\langle 22, 8 \rangle$ | -500 (incl. float) | Control (RL) |
| | MLP-AC-L | [6,32,32,4] | ReLU | 1412 | – | – | 1e$-$4 | $\langle 22, 8 \rangle$ | -500 (incl. float) | Control (RL) |
| | MLP-DQN | [6,128,128,3] | ReLU | 17,795 | – | – | 1e$-$6 | float | $-298$ | Control (RL) |

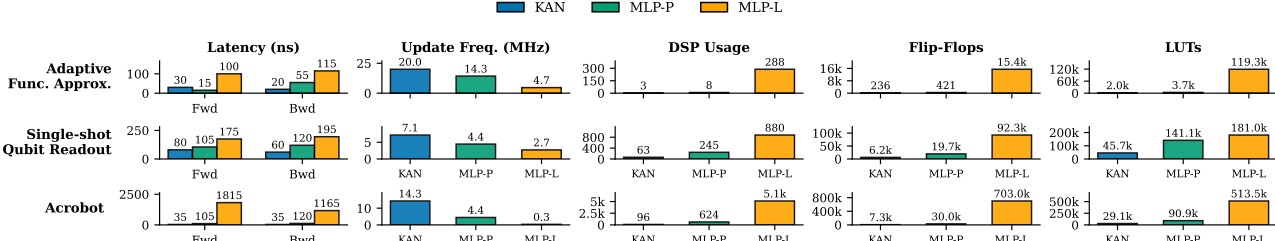

*Figure 3.* **Post-synthesis FPGA latency and resource cost for fully online learning.** Models are synthesized for an AMD Virtex™ UltraScale+™ XCVU13P at 200 MHz. Across tasks, KAN achieves the best latency-resource trade-off (DSP/FF/LUT) and the highest update rate (online updates/s, forward + backward). MLPs use more resources and/or diverge under fixed-point updates. No BRAM was used for either architecture across these experiments. Full post-route place and validation is detailed in Appendix B.8.

fixed-point constraints, while neither LayerNorm nor data-dependent initialization methods such as LSUV (Mishkin & Matas, 2016b) close the performance gap (see Appendix D). MLPs also broadly encompass methods such as LMS filters, ELMs, and related approaches. To justify this design choice, we evaluate Online Sequential Extreme Learning Machines (OS-ELMs) on the complex qubit readout task in Table 1. Even with floating-point training, OS-ELMs perform poorly, suggesting that these non-gradient-trained shallow models lack the expressivity required for the learning settings considered here. Temporal architectures like SNNs represent a distinct class of methods due to non-deterministic latency characteristics.

**Metrics.** We report task-appropriate online performance metrics: cumulative regret for drifting regression, running accuracy for qubit readout, and episodic return for control. We also report post-synthesis FPGA latency and resource utilization (LUTs/FFs/BRAMs/DSPs), which match full post-route implementations and confirm they successfully meet timing constraints (Appendix B.8).

**Hyperparameters.** Architectures, learning rates, and fixed-point formats are summarized in Table 1.

### 5.2. Benchmark 1: Adaptive Function Approximation Under Concept Drift

We first consider a controlled non-stationary regression task as a proxy for real-time sensor calibration and signal recovery, where the underlying transfer function changes and must be tracked online.

**Task.** The task runs for $T = 1500$ steps. Inputs $x_t \sim \mathcal{U}[-1, 1]$ and targets follow a piecewise latent function with regime changes at $t = 500$ and $t = 1000$:

$$y_t = \begin{cases} \sin(x_t) + 0.3x_t^2 & 0 \leq t < 500 \\ -\cos(2x_t) + 0.1x_t^3 + 1.0 & 500 \leq t < 1000 \\ \exp(-0.5(x_t - 1)^2) + 0.05x_t^3 & 1000 \leq t < 1500 \,. \end{cases}$$

**Protocol and metric.** Models are updated online with batch size 1 using a fixed learning rate $\eta$ and the instantaneous gradient of the squared error. We report cumulative regret $R_T = \sum_{t=1}^{T} \text{MSE}_t$ to capture both convergence speed and adaptation after regime shifts.

**Results.** Figure 4 shows that in floating point, KAN achieves the lowest cumulative regret and re-adapts immediately after each regime change, while the parameter-matched MLP-P fails to track the drift and its regret grows steadily. A larger MLP-L can eventually recover, but adapts more

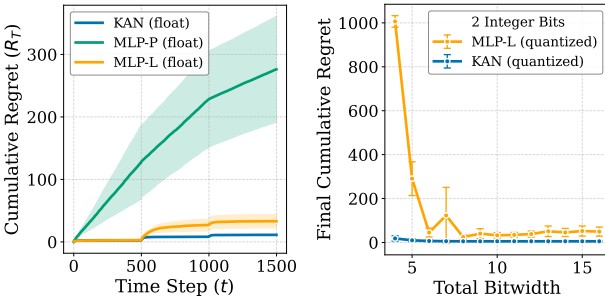

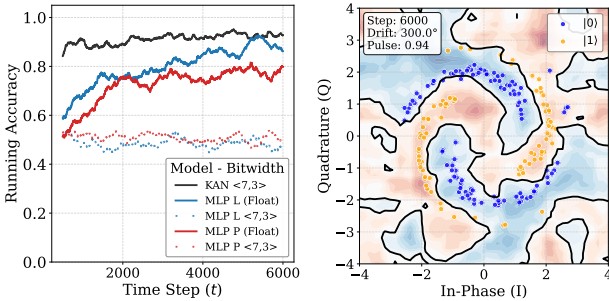

*Figure 4.* **Adaptive function approximation under concept drift.** **Left:** Cumulative regret with regime changes at $t = 500, 1000$: KAN adapts rapidly, MLP-P diverges, and MLP-L converges more slowly. **Right:** Final cumulative regret vs. fixed-point bitwidth (2 integer bits): KAN remains stable under quantization while MLP-L degrades at low precision.

*Figure 6.* **Adaptive single-shot qubit readout.** **Left:** Online running accuracy. Quantized KAN tracks drift and avoids the collapse observed in quantized MLPs. **Right:** Snapshot of learned decision boundary at $t = 6000$: KAN tracks the drifting IQ distributions despite large phase drift.

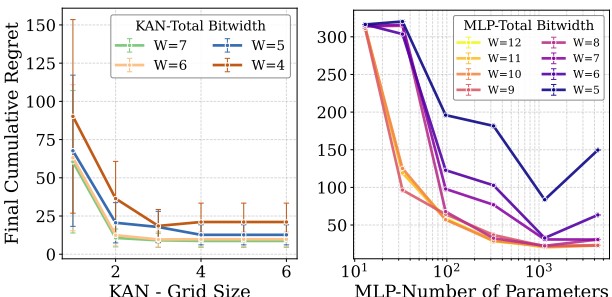

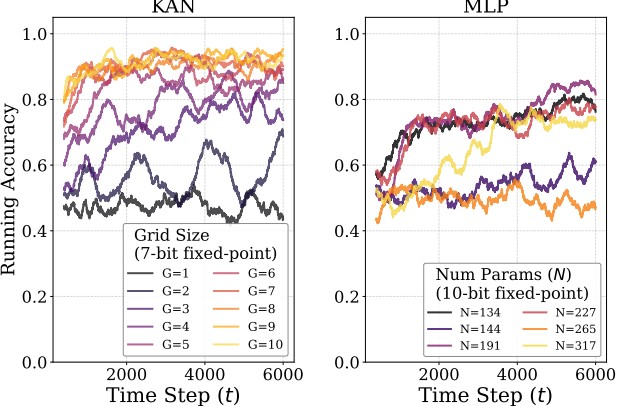

*Figure 5.* **Empirical validation of approximation under quantization.** **Left (KAN):** Error improves with grid size $G$ until reaching a quantization floor set by bitwidth. **Right (MLP):** Increasing parameter count $N$ amplifies sensitivity to quantization and destabilizes learning.

*Figure 7.* **Capacity-precision decoupling under drift.** **Left (KAN):** Accuracy improves monotonically with grid size $G$ even under aggressive quantization. **Right (MLP):** Under 10-bit fixed-point training, increasing parameter count $N$ yields only marginal accuracy gains and often destabilizes learning.

slowly after shifts, yielding higher regret over the stream. Under fixed-point quantization (2 integer bits), KAN remains stable across total bitwidths, with only mild degradation at very low precision. In contrast, MLP-L exhibits a sharp precision cliff: low bitwidth leads to large final regret and high variance, and increasing bitwidth is required to regain stable learning. Figure 5 further shows that KAN benefits from increasing grid size $G$ until reaching a quantization-limited floor set by bitwidth, whereas for MLPs, increasing parameter count $N$ shows diminishing returns and amplifies sensitivity to quantization, destabilizing learning at low precision.

### 5.3. Benchmark 2: Adaptive Single-shot Qubit Readout

FPGAs are central to quantum computing platforms for deterministic, ultra-low-latency readout, feedback, and calibration (Guglielmo et al., 2025; Berritta et al., 2024). We cast single-shot qubit readout as an online binary classification problem with drifting, non-linearly separable decision boundaries.

**Task.** Each time step yields a noisy IQ sample $\mathbf{x}_t = (I_t, Q_t)$ from a rotating XOR constellation with Gaussian noise, Kerr-type phase distortion, and slow global drift. Full task details are provided in Appendix C.

**Protocol and metric.** Models are updated online from local label feedback at each step, with no buffering. We report the instantaneous accuracy and its running average over time.

**Results.** Figure 6 illustrates fully online running accuracy under drifting, non-linearly separable IQ constellations and visualizes the learned decision boundary at a representative time. Figure 7 sweeps capacity at fixed precision, KAN grid size $G$ versus MLP parameter count $N$. KAN performance improves monotonically with larger $G$, whereas for MLPs increasing $N$ is the only scaling knob and often destabilizes convergence under fixed-bit arithmetic. Figure 1 quantifies how our synthesized fixed-point online training kernels scale with trainable parameters.

## 5.4. Benchmark 3: Online Policy Optimization under Non-stationary Dynamics

We evaluate the agent's ability to adapt to non-stationary physical dynamics on `Acrobot-v1` (Sutton, 1995).

**Task.** The agent observes a 6D state and selects one of three torques. To induce non-stationarity, we randomize the link masses ($m \in [0.8, 1.2]$) and lengths ($l \in [0.9, 1.1]$) at the start of every episode, requiring the agent to continually adjust its control policy.

**Online actor-critic.** We use an *online N-step actor-critic* update ($N=3$) with a shared network that outputs (i) action logits and (ii) a state-value estimate. At each step, we sample an action from the softmax policy and perform a single batch update using an $N$-step bootstrapped return. This setting is deliberately stringent: non-stationary dynamics, stochastic exploration, and fully online updates amplify gradient interference in dense networks.

**Baselines and metric.** We compare (i) our fixed-point KAN policy/value network to (ii) MLP baselines (small/large, float) trained with *identical* online actor-critic rule, and (iii) a streaming value-learning baseline: online DQN-style Q-learning ($\epsilon$-greedy, Huber TD loss) without replay. We report episodic return over time (mean $\pm$ std over random seeds) and mark the environment's solved threshold.

**Results.** As shown in Figure 8, under per-episode randomization, KAN sustains online improvement with fixed-point updates, while actor-critic MLPs are markedly less stable even in floating point. A larger MLP can recover only with the more stable value-learning update, supporting our claim that *update sparsity mitigates interference in non-stationary streams* (Lemma 3.1).

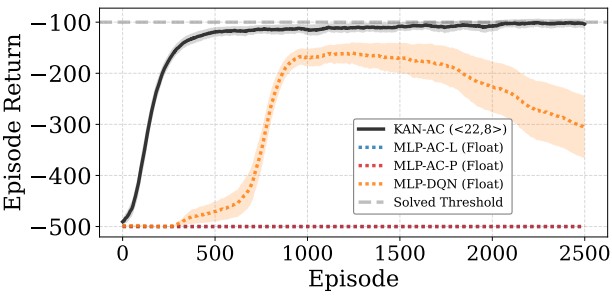

*Figure 8.* **Non-stationary Acrobot control.** Episodic return under randomized dynamics (link masses/lengths resampled each episode; mean±std over seeds). The fixed-point KAN actor-critic (black) reaches the solved regime in ∼300 episodes and remains stable, while online actor-critic MLPs fail to adapt even at 10× the KAN parameter budget. An MLP only succeeds with a more stable online DQN update and a much larger network (∼120×), but learns more slowly and degrades under continued randomization.

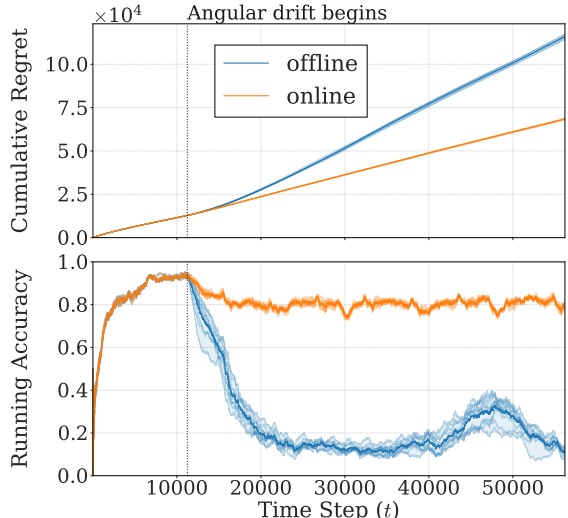

*Figure 9.* **Comparison of online and offline learner performance on digit classification with angular drift.** The online learner maintains an accuracy of 80% even with extreme angular drifts. In comparison, the frozen learner is only able to consistently classify around 20% of inputs correctly, with a cumulative regret over 50% that of the online learner.

## 5.5. Hardware results

We report post-synthesis FPGA latency and resource utilization for all models in Figure 3. B-spline locality yields sparse KAN updates, reducing DSP/FF/LUT usage and latency versus a parameter-matched MLP (Proposition 3.2), while keeping both forward and backward passes in the sub-100ns regime across all experiments. To ensure these HLS estimates reflect physically realizable hardware, we also performed full post-route implementations for the primary KAN designs, confirming they successfully meet timing constraints and achieve sub-microsecond latency (Appendix B.8).

## 6. Scalability

The preceding benchmarks already establish the key comparative conclusions against dense MLPs: (i) locality yields sparse per-sample updates and lower update cost (Lemma 3.1, Proposition 3.2), and (ii) capacity can be increased via grid resolution $G$ without increasing the active compute per sample (Section 3.2), while maintaining stability under fixed-point updates (Propositions 3.3–3.4). In this section, we thus focus on the scaling properties of online KANs through higher-dimensional tasks.

### 6.1. Digit Classification with Online Drift

We take the classic problem of digit classification and introduce a constant *angular drift*, or rotation, over time. This drift is representative of many types of sensor degradations

and distribution shifts that occur in real-time systems. By the final timestep, images have rotated nearly $230°$, making rapid adaptation essential yet tricky for model-free online learners.

**Task Setup.** We use the UCI dataset containing 5,620 flattened $8 \times 8$ images of handwritten digits (Alpaydin & Kaynak, 1998). The online learner is initially fed in $N_{\text{stationary}} = 2$ epochs of images; after this, we train for $N_{\text{rotating}} = 8$ epochs where the image on step $t$ is rotated by $\theta(t) = \omega t$, with $\omega = 0.005°/\text{step}$.

**Evaluation.** To evaluate adaptation speed, we plot cumulative regret on the cross-entropy loss as well as running accuracy on the past 1,000 samples. To demonstrate the necessity of online learning, we compare to a baseline learner that does not update its parameters for the rotating epochs.

**Results.** Figure 9 demonstrates that the online learner is significantly more accurate and stable, demonstrating its ability to adapt to extreme distribution shifts when compared to a pretrained model.

### 6.2. Ablation Studies

We study scalability to higher-dimensional inputs by taking the architecture used for the digits task and scaling up input dimension, freezing all other hyperparameters. As shown in Figure 10, we maintain sub-$\mu$s forward *and* backward passes even for models exceeding $10^5$ trainable parameters. Because KAN layers are more expressive than MLPs, they typically require less depth to achieve comparable performance; paired with sparse, locality-driven gradient updates, this enables ultrafast backpropagation at scales difficult to attain with on-chip dense MLP training.

## 7. Limitations

We use fully online, single batch SGD update to meet sub-$\mu$s streaming constraints; this differs from common RL practice (e.g. PPO), which leverages trajectory batches and can converge faster. While our Acrobot setting is intentionally stringent (per-episode randomization and stochastic exploration), our kernels could be extended to accumulate gradients over short windows, trading latency and resources for stability while remaining streaming-friendly. More work also needs to be done in understanding how other optimizers (e.g. Adam) work under fixed-point learning dynamics for KAN and MLP architectures.

For scalability, the reported ablations assume a fixed set of hyperparameters; in other regimes, architectural choices may affect resource usage and latency, although we expect linear scaling trends to persist. For example, higher task complexity may necessitate increasing the spline order, which was previously held constant, to maintain sufficient

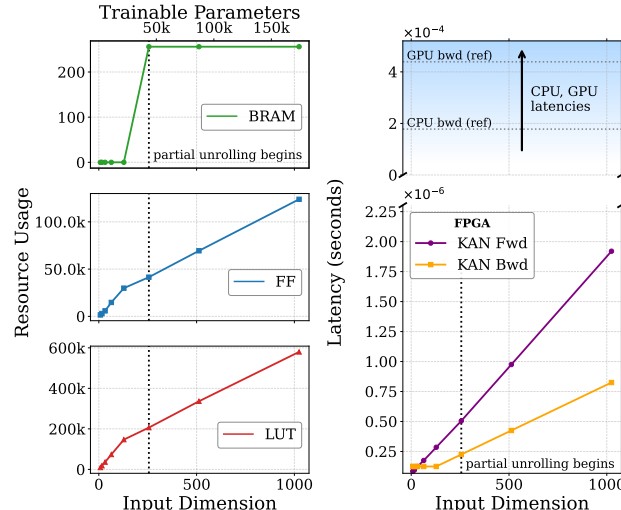

*Figure 10.* **Resource and latency scaling estimates with increasing input dimension.** Resource usage and latency scales roughly linearly with the input dimension and thus the trainable parameters. We partially unroll (reduce parallelism) for input dimensions greater than 256 to accommodate larger networks on the FPGA. Exact latency and resource values depend on specific implementation techniques, which can be tuned for the desired application.

representational capacity.

Additionally, while we use HLS to facilitate rapid prototyping and broad design-space exploration, manual RTL refinement could further improve hardware efficiency. To ensure that HLS estimates do not overstate our latency claims, we performed full post-route implementation for the main KAN designs (Appendix B.8), all of which met timing with positive slack and remained well below 1 $\mu$s end-to-end. Overall, our focus remains on the architectural merits of KANs and order-of-magnitude estimates and comparisons, rather than extreme, low-level hardware optimization.

## 8. Conclusion

This work shows that Kolmogorov–Arnold Networks (KANs) are uniquely suited to ultrafast on-chip online learning, where MLPs fundamentally struggle. By exploiting B-spline locality, KANs replace dense, interference-prone gradient updates with sparse, targeted coefficient adjustments, while scaling capacity via grid size at near-constant per-sample cost. Additionally, KANs possess theoretical properties that make them inherently robust to fixed-point quantization.

Realized as a custom-hardware FPGA implementation, KANs enable favorable resource scaling and sub-microsecond forward and backward passes with fully on-chip parameter updates, which are three to four orders of magnitude faster than host–accelerator alternatives. Across

experiments with drifting regression, adaptive qubit readout, and non-stationary control, KANs consistently outperform parameter-matched MLPs that either diverge under quantization or require impractical precision to stabilize.

More broadly, KANs' core hardware advantages remain underexplored because modern ML stacks, and GPUs in particular, are built around dense, batched linear algebra rather than localized basis-function learning. Hardware that directly supports spline evaluation and basis selection would make KAN inference and sparse gradient updates hardware-native, enabling ultrafast adaptive control for emerging applications including quantum and plasma systems, networking, and more.

## Acknowledgements

DH, AG, and PH are supported by the NSF-funded A3D3 Institute (NSF-PHY-2117997).

## Impact Statement

This paper presents work whose goal is to advance the field of Machine Learning. There are many potential societal consequences of our work, none of which we feel must be specifically highlighted here.

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

# A. Proofs of Propositions

## A.1. Proof of Update Complexity (Proposition 3.2)

**Setup:** Let the total parameter budget be $N$. We compare a standard MLP layer with input dimension $d_{\text{in}}^{\text{MLP}}$ and output dimension $d_{\text{out}}^{\text{MLP}}$ against a KAN layer with input and output dimensions $d_{\text{in}}^{\text{KAN}}$ and $d_{\text{out}}^{\text{KAN}}$. Let $S$ denote the B-spline order and $G$ denote the grid size. For the MLP layer, the total number of parameters is $N_{\text{MLP}} = d_{\text{in}}^{\text{MLP}} d_{\text{out}}^{\text{MLP}}$ while for the KAN layer it is $N_{\text{KAN}} = d_{\text{in}}^{\text{KAN}} d_{\text{out}}^{\text{KAN}} (G + S)$. To maintain the fixed budget $N = N_{\text{MLP}} = N_{\text{KAN}}$, we assume $d_{\text{in}}^{\text{MLP}} d_{\text{out}}^{\text{MLP}} = N$ and $d_{\text{in}}^{\text{KAN}} d_{\text{out}}^{\text{KAN}} = N/(G + S)$.

*Proof.* We analyze the scalar arithmetic operations required to compute the gradient with respect to the weights for a single sample $(x, y)$. Let $\vec{g} \in \mathbb{R}^{d_{\text{out}}}$ be the backpropagated error signal (gradient of loss w.r.t layer output).

**Case 1: MLP Update Complexity.** The gradient update for an MLP weight matrix $W \in \mathbb{R}^{d_{\text{out}} \times d_{\text{in}}}$ is given by the outer product:

$$\nabla_W \mathcal{L} = \vec{g} x^{\top}$$

For each element $W_{q,p}$, the computation requires 1 multiplication. The total number of update operations is:

$$\mathcal{C}_{\text{MLP}} = d_{\text{in}}^{\text{MLP}} d_{\text{out}}^{\text{MLP}} = N$$

**Case 2: KAN Update Complexity.** The KAN layer output is $y_q = \sum_{p=1}^{d_{\text{in}}} \sum_{i=1}^{G+S} w_{q,p,i} B_i(x_p)$. The gradient with respect to a specific spline coefficient $w_{q,p,i}$ is:

$$\frac{\partial \mathcal{L}}{\partial w_{q,p,i}} = \frac{\partial \mathcal{L}}{\partial y_q} \frac{\partial y_q}{\partial w_{q,p,i}} = g_q B_i(x_p)$$

Due to the local support property of B-splines (Lemma 3.1), for a given input coordinate $x_p$, $B_i(x_p)$ is nonzero for exactly $S + 1$ indices. Let $\mathcal{K}_p = \{i \mid B_i(x_p) \neq 0\}$ be the set of active indices for input $p$, where $|\mathcal{K}_p| = S + 1$. The gradient is zero for all $i \notin \mathcal{K}_p$. Therefore, we only perform updates for indices in $\mathcal{K}_p$.

The total number of operations is the sum of updates over all edges $(p, q)$:

$$\mathcal{C}_{\text{KAN}} = \sum_{q=1}^{d_{\text{out}}} \sum_{p=1}^{d_{\text{in}}} |\mathcal{K}_p| = d_{\text{out}} d_{\text{in}} (S + 1)$$

Substituting the parameter count relationship $d_{\text{in}}^{\text{KAN}} d_{\text{out}}^{\text{KAN}} = N/(G + S)$:

$$\mathcal{C}_{\text{KAN}} = \frac{N}{G + S} \cdot (S + 1) = N \left( \frac{S + 1}{G + S} \right)$$

**Conclusion.** Comparing the two complexities:

$$\frac{\mathcal{C}_{\text{KAN}}}{\mathcal{C}_{\text{MLP}}} = \frac{N((S + 1)/(G + S))}{N} = \frac{S + 1}{G + S}$$

Since $S$ is fixed and $G$ scales with grid resolution (typically $G \gg S$), the KAN update complexity is strictly sublinear relative to the parameter budget. $\qquad\square$

## A.2. Proof of KAN Activation Bounds (Proposition 3.3)

*Proof.* Let $m = \min_i W_i$ and $M = \max_i W_i$. For normalized B-splines, we have for any $x$ in the domain:

$$B_i(x) \geq 0 \quad \text{and} \quad \sum_i B_i(x) = 1.$$

Consider

$$\phi(x) = \sum_i W_i B_i(x).$$

Using the partition of unity,

$$\phi(x) - m = \sum_i W_i B_i(x) - m \sum_i B_i(x) = \sum_i (W_i - m) B_i(x) \geq 0,$$

since $W_i - m \geq 0$ and $B_i(x) \geq 0$. Thus $\phi(x) \geq m$.

Similarly,

$$M - \phi(x) = \sum_i M B_i(x) - \sum_i W_i B_i(x) = \sum_i (M - W_i) B_i(x) \geq 0,$$

so $\phi(x) \leq M$.

Hence,

$$\min_i W_i \leq \phi(x) \leq \max_i W_i.$$

$\square$

### A.3. Proof of Bounded Gradient Sensitivity (Proposition 3.4)

*Proof.* We examine the gradient of the loss $\mathcal{L}$ with respect to parameters in a single layer $\ell$, considering a connection from input node $p$ to output node $q$. Let $g_{\ell+1,q} = \frac{\partial \mathcal{L}}{\partial x_{\ell+1,q}}$ denote the backpropagated gradient.

**Case 1: MLP.** The pre-activation output is

$$x_{\ell+1,q} = \sum_{p=1}^{d_{\text{in}}} W_{\ell,q,p}\, x_{\ell,p}.$$

By the chain rule, the gradient with respect to $W_{\ell,q,p}$ is

$$\frac{\partial \mathcal{L}}{\partial W_{\ell,q,p}} = g_{\ell+1,q} \cdot \frac{\partial x_{\ell+1,q}}{\partial W_{\ell,q,p}} = g_{\ell+1,q} \cdot x_{\ell,p}.$$

Critically, the gradient magnitude scales with $|x_{\ell,p}|$, coupling the update dynamic range to the input distribution.

**Case 2: KAN.** The layer output is

$$x_{\ell+1,q} = \sum_{p=1}^{n_\ell} \phi_{\ell,q,p}(x_{\ell,p}) = \sum_{p=1}^{n_\ell} \sum_{i=1}^{G+S} w_{\ell,q,p,i}\, B_i(x_{\ell,p}).$$

The gradient with respect to coefficient $w_{\ell,q,p,i}$ is

$$\frac{\partial \mathcal{L}}{\partial w_{\ell,q,p,i}} = g_{\ell+1,q} \cdot \frac{\partial x_{\ell+1,q}}{\partial w_{\ell,q,p,i}} = g_{\ell+1,q} \cdot B_i(x_{\ell,p}).$$

By the partition of unity property of B-splines, $\sum_i B_i(x) = 1$ and $0 \leq B_i(x) \leq 1$ for all $x$ and $i$. Therefore,

$$\left| \frac{\partial \mathcal{L}}{\partial w_{\ell,q,p,i}} \right| = |g_{\ell+1,q}| \cdot |B_i(x_{\ell,p})| \leq |g_{\ell+1,q}|.$$

Unlike the MLP case, the gradient dynamic range is completely decoupled from the input magnitude $|x_{\ell,p}|$ as $B_i(x) \in [0,1]$.

$\square$

## B. FPGA Implementation Details

### B.1. Code Generation and Build Flow

We generate one self-contained Vitis HLS project per model configuration. A lightweight Python frontend writes a set of C++ headers/sources from templates: (i) `defines.h` (dimensions, precisions, grid constants, learning rate), (ii) `parameters.h` (parameter and context structs, plus parameter initialization), (iii) `components.h` (layer primitives: forward/backward/update), and (iv) a top-level kernel (`kan.cpp` / `mlp.cpp`) that invokes all layers in sequence. A `build.tcl` script specifies the target FPGA part and clock period for Vitis HLS synthesis.

## B.2. Fixed-Point Formats and Saturation

All arithmetic uses either `ap_fixed⟨W, I⟩` with rounding and saturation (AP_RND_CONV, AP_SAT) or IEEE float (for ablations). The float datatype is used only to study convergence behavior, not for hardware analysis. We define three independent types: `input_t` for inputs, `weight_t` for parameters/accumulators, and `output_t` for outputs/feedback. The learning rate is compiled as a constant of type `weight_t`.

## B.3. KAN Layer Implementation

**Uniform grid and indexing.** Let the grid cover $[x_{\min}, x_{\max}]$ with $G$ cells of width $H = (x_{\max} - x_{\min})/G$. For each coordinate $x$, we compute $t = (x - x_{\min})/H$, clamp $t$ to $[0, G)$, and set $k = \lfloor t \rfloor$ (cell index). Within the cell, we use a small fractional index $u$ obtained from the low bits of a fixed-point representation of $t$, producing `LUT_RESOLUTION`$= 2^F$ bins per cell. We store $(k, u)$ for each $(q, p)$ pair in a per-layer context to make the backward pass purely index-driven.

**Basis LUTs (values and derivatives).** For spline order $S$, exactly $S + 1$ basis functions are nonzero per cell. We precompute a constant LUT:

$$\texttt{LUT.B}_r[u] \approx B_r(\xi_u), \qquad \texttt{LUT.dB}_r[u] \approx \frac{d}{dx}B_r(\xi_u),$$

for $r = 0, \ldots, S$ and $u = 0, \ldots, 2^F - 1$, where $\xi_u$ are uniformly spaced sample points over a unit cell. $F$ is a hyperparameter for how many bits are used to address `LUT.B`$_r[u]$ and is specified in the user-defined configuration The LUT is compiled into `parameters.h` as a read-only constant and bound to on-chip ROM (LUTRAM).

**Forward pass (KAN).** Each layer stores parameters as `Ws[q][p][c]` with coefficient index $c \in \{0, \ldots, (G+S-1)\}$. Given $(k, u)$ for input $x_p$, we evaluate only the active coefficients:

$$\phi_{q,p}(x_p) \approx \sum_{r=0}^{S} \texttt{Ws}[q][p][k + r] \cdot \texttt{LUT.B}_r[u], \qquad y_q = \sum_p \phi_{q,p}(x_p).$$

In hardware, the inner sums are fully unrolled across $r$ and typically across $p$ (subject to resource constraints), with a final unrolled reduction over $p$.

**Backward pass and in-place update (KAN).** For the upstream gradient $g_q = dL/dy_q$, the coefficient update for the active cell is

$$\texttt{Ws}[q][p][k + r] \leftarrow \texttt{Ws}[q][p][k + r] - \eta\, g_q\, \texttt{LUT.B}_r[u], \quad r = 0, \ldots, S.$$

The downstream gradient for each input coordinate is computed via LUT derivatives:

$$\frac{dL}{dx_p} = \sum_q g_q \sum_{r=0}^{S} \texttt{Ws}[q][p][k + r] \cdot \texttt{LUT.dB}_r[u].$$

In the implementation, $(k, u)$ are read from context, the $r$-loop is unrolled, and $dL/dx_p$ is accumulated across $q$.

## B.4. MLP Baseline Implementation

**Forward.** Each layer computes $z_q = \sum_p W[q][p]x_p + b[q]$ and applies a hardware-friendly activation. In our experiments, we use ReLU; we also support piecewise `hard_tanh` and `hard_silu` variants. For backprop, we store in context: (i) a copy of the layer input $x$ and (ii) the pre-activation vector $z$ (except for the final linear layer).

**Backward and update.** Given upstream gradient $dL/dy$, we compute $dL/dz = (dL/dy) \odot f'(z)$ for hidden layers (and $dL/dz = dL/dy$ for the linear output layer), then apply in-place SGD updates:

$$b[q] \leftarrow b[q] - \eta\, dL/dz_q, \qquad W[q][i] \leftarrow W[q][i] - \eta\, dL/dz_q\, x_i.$$

The downstream gradient is

$$\frac{dL}{dx_p} = \sum_q (dL/dz_q)\, W_{\text{old}}[q][p],$$

where we use $W_{\text{old}}$ (the weight prior to update) to match standard backprop ordering.

**Initialization and parity with PyTorch.** Weights and biases are initialized with the same per-layer scale as common PyTorch uniform schemes (using a scale proportional to $1/\sqrt{d_{\text{in}}}$) to ensure controlled comparisons.

## B.5. HLS Directives and On-Chip Storage

**Deterministic control.** All loops have static bounds determined by compile-time dimensions. The kernels are structured to enable deterministic scheduling: unrolled inner loops for parallel MACs/coefficient updates and pipelining across output neurons where appropriate.

**LUT storage.** KAN basis/derivative LUTs are bound to single-port ROM implemented in LUTRAM (read-only, constant at synthesis), minimizing BRAM pressure.

**Parameter storage.** KAN parameters are stored as a 3D array `Ws[q][p][c]`. We apply complete partitioning across `q` and `p`, and cyclic partitioning across the coefficient dimension `c` with factor $(S+1)$ to match the number of active basis functions per cell. MLP weights `W[q][p]` and biases `b[q]` are partitioned to expose full parallelism across dot-products and updates. Per-layer contexts (KAN indices; MLP cached $x$ and $z$) are fully partitioned and reside in registers/small RAMs.

## B.6. Software Interface and Verification

**CPU-loadable functional model.** For rapid functional testing and simulation of fixed-point arithmetic, we compile the generated HLS C++ into a CPU-loadable shared library and invoke it from Python via `ctypes`. The exposed interface accepts pointers to the input, output, and feedback tensors, together with a `zero_grad` flag controlling whether parameter updates are applied. This setup enables fast iteration and direct comparison against software reference implementations without FPGA synthesis.

**Functional verification.** We use the shared library to validate both numerical correctness and learning dynamics. For each architecture, we first verify convergence behavior in floating-point mode using HLS emulation. In this setting, the HLS MLP implementation matches an equivalent PyTorch model exactly, including initialization and update ordering, providing a strong software reference. All fixed-point experiments are then performed with identical architectures and learning rules, ensuring that any observed differences arise solely from quantization and hardware execution effects.

## B.7. HLS Synthesis

The verified HLS designs are synthesized using Vitis HLS via an automatically generated `build.tcl` script. Synthesis reports provide estimates of kernel latency, initiation interval, and on-chip resource utilization (LUTs, DSPs, BRAMs). Unless otherwise stated, all designs target an AMD Virtex™ UltraScale+™ XCVU13P FPGA and are synthesized with a target clock period of 5 ns (200 MHz). All comparisons between KAN and MLP implementations are performed under identical timing, precision, and synthesis constraints.

## B.8. Post-Route Implementation Validation

Although the main hardware results are reported from Vitis HLS synthesis to enable broad design-space exploration across bitwidths, architectures, and tasks, we further validate the physical realizability of the KAN latency claims by running full implementation on the target FPGA fabric. We used Vivado 2024.1 with the Vitis HLS `export_design -format ip_catalog -flow impl` workflow, targeting the AMD Virtex UltraScale+ XCVU13P device and a 5 ns clock constraint. This flow performs logic synthesis, placement, routing, and post-route timing analysis.

Because our KAN kernels have static loop bounds, deterministic control flow, fully on-chip state, and no external DRAM accesses, their latency in clock cycles is fixed by the HLS schedule. The primary risk to sub-$\mu$s operation is therefore failure to meet the target clock period after routing. As shown in Table 2, all implemented KAN designs meet the 5 ns timing constraint with positive worst negative slack (WNS), validating that the reported forward-plus-backward latencies are physically realizable on the target FPGA.

*Table 2.* **Post-route validation of KAN latency on AMD Virtex UltraScale+ XCVU13P.** All designs target a 5 ns clock period. Latency denotes one complete online update, including forward pass, backward pass, and in-place parameter update. *Note: Total Latency = Forward + Backward pass.*

| Task | Achieved Clock (ns) | WNS (ns) | Latency (ns) | Post-Route LUT/FF/DSP |
|---|---|---|---|---|
| Adaptive Function Approximation | 3.042 | +1.958 | 50 | 430 / 199 / 4 |
| Single-Shot Qubit Readout | 3.871 | +1.129 | 140 | 10,592 / 3,858 / 84 |
| Non-Stationary Acrobot Control | 3.884 | +1.116 | 70 | 7,666 / 4,385 / 106 |

## C. Benchmark 2 Details: Adaptive Single-shot Qubit Readout

We instantiate adaptive single-shot readout as a *fully online* binary classification problem on streaming IQ samples. At each time step $t$, the learner receives a single point $\mathbf{x}_t = (I_t, Q_t) \in \mathbb{R}^2$, predicts $\hat{y}_t \in \{-1, +1\}$, and immediately receives the true label $y_t \in \{-1, +1\}$ and updates once. The data distribution drifts continuously in time and is intentionally non-linearly separable due to a Kerr-type phase distortion.

### C.1. Rotating XOR constellation with Kerr distortion and drift

**Latent state and class definition.** Each sample is generated by first drawing a latent "computational-basis" index

$$s_t \sim \text{Unif}\{0, 1, 2, 3\},$$

corresponding to one of four Gaussian blobs centered in the four quadrants of the IQ plane. We use an XOR/parity labeling:

$$y_t = \begin{cases} -1, & s_t \in \{0, 1\} \quad \text{(quadrants 1 \& 3)} \\ +1, & s_t \in \{2, 3\} \quad \text{(quadrants 2 \& 4).} \end{cases}$$

This creates the classic 2D XOR decision boundary (non-linearly separable).

**Base blob centers and noise.** Let $\texttt{spread} > 0$ set the separation between blob centers and $\sigma > 0$ set isotropic Gaussian noise. We define the unrotated centers

$$\boldsymbol{\mu}_0 = (+\texttt{spread}, +\texttt{spread}), \quad \boldsymbol{\mu}_1 = (-\texttt{spread}, -\texttt{spread}),$$
$$\boldsymbol{\mu}_2 = (-\texttt{spread}, +\texttt{spread}), \quad \boldsymbol{\mu}_3 = (+\texttt{spread}, -\texttt{spread}).$$

then sample

$$(I, Q) \sim \mathcal{N}(\boldsymbol{\mu}_{s_t}, \sigma^2 \mathbf{I}_2).$$

**Kerr-type phase distortion.** To emulate non-linear IQ warping (e.g. due to Kerr physics), we apply a phase twist that grows with amplitude. Let $r = \sqrt{I^2 + Q^2}$ and $\phi = \text{atan2}(Q, I)$. We compute

$$\phi' = \phi + \kappa r^2, \qquad (I, Q) \leftarrow (r \cos \phi', \ r \sin \phi'),$$

where $\kappa$ controls distortion strength. This transforms circular blobs into "comet-shaped" clusters and makes simple linear or distance-based boundaries suboptimal.

---

**Algorithm 1** Rotating XOR stream with Kerr distortion

---

1: Sample latent index $s \sim \text{Unif}\{0, 1, 2, 3\}$ and set label $y \in \{-1, +1\}$ by parity.
2: Set center $\boldsymbol{\mu}_s \in \mathbb{R}^2$ from the quadrant list.
3: Sample $(I, Q) \sim \mathcal{N}(\boldsymbol{\mu}_s, \sigma^2 \mathbf{I}_2)$.
4: Compute $r = \sqrt{I^2 + Q^2}$, $\phi = \text{atan2}(Q, I)$, and apply Kerr twist $\phi \leftarrow \phi + \kappa r^2$.
5: Set $(I, Q) \leftarrow (r \cos \phi, \ r \sin \phi)$.
6: Apply breathing $(I, Q) \leftarrow p_t(I, Q)$ with $p_t = 1 + a \sin(\omega t)$.
7: Rotate by $\theta_t = \text{rad}(t \cdot \texttt{drift\_speed})$: $(I, Q) \leftarrow R(\theta_t)(I, Q)$.
**output** $\mathbf{x}_t = (I, Q)$ and $y_t = y$.

---

*Table 3.* Default stream parameters for Benchmark 2 (matching the code).

| Parameter | Symbol | Value |
|---|---|---|
| Blob separation | `spread` | 1.5 |
| Gaussian noise std. | $\sigma$ (`noise_scale`) | 0.4 |
| Kerr distortion strength | $\kappa$ (`kerr_strength`) | 0.4 |
| Rotation rate (deg/step) | `drift_speed` | 0.05 |
| Breathing amplitude | $a$ | 0.2 |
| Breathing frequency | $\omega$ | 0.01 |

**Slow drift: global rotation and breathing.** We impose two forms of non-stationarity.

*(i) Global rotation.* At time $t$, we rotate the entire constellation by angle

$$\theta_t = \mathrm{rad}(t \cdot \texttt{drift\_speed}),$$

where $\mathrm{rad}(\cdot)$ converts degrees to radians (matching the implementation). We then apply

$$\begin{bmatrix} I \\ Q \end{bmatrix} \leftarrow \begin{bmatrix} \cos\theta_t & -\sin\theta_t \\ \sin\theta_t & \cos\theta_t \end{bmatrix} \begin{bmatrix} I \\ Q \end{bmatrix}.$$

*(ii) Breathing/pulsing.* We optionally modulate the radius by a slow sinusoid

$$(I, Q) \leftarrow p_t(I, Q), \qquad p_t = 1 + a\sin(\omega t),$$

which induces gradual expansion/contraction and further stresses online adaptation.

### C.2. Reference implementation and hyperparameters

Algorithm 1 summarizes the generator used in our experiments. Unless stated otherwise, we use the default parameters in Table 3, which match the provided code.

### C.3. Online protocol and reported metric

At each step $t$, the learner: (i) observes $\mathbf{x}_t$, (ii) outputs $\hat{y}_t \in \{-1, +1\}$, (iii) receives $y_t$ immediately, and (iv) performs a single gradient update using only the current sample (no replay/buffering). We report *instantaneous accuracy* $\mathbf{1}\{\hat{y}_t = y_t\}$ and its running mean over time (the plotted "running accuracy"). When visualizing decision boundaries (Fig. 6), we freeze model parameters at the indicated time and evaluate $\hat{y}$ on a dense grid in the $(I, Q)$ plane.

**Reproducibility.** All results are generated using the same streaming procedure described above, with fixed seeds per run. When sweeping capacity (KAN grid size $G$ or MLP parameter count $N$), we keep the stream parameters fixed (Table 3) and vary only model/hardware configurations.

## D. Extended MLP Baselines: Initialization and Normalization

To demonstrate KAN's advantage against a stronger Multi-Layer Perceptron (MLP) baseline and to quantify the hardware trade-offs of common software stabilization techniques, we augmented our MLP baselines in two ways:

- **MLP + LSUV Init**: We apply the data-dependent initialization scheme of (Mishkin & Matas, 2016a) to the MLP weights before online learning begins. This gives the MLP a calibrated starting point without adding any runtime overhead during inference or training.

- **MLP + LayerNorm**: We added a LayerNorm (Ba et al., 2016) layer after each hidden layer. We used High-Level Synthesis (HLS) to implement this entirely in fixed-point arithmetic with trainable affine parameters ($\gamma$, $\beta$). The forward pass computes per-layer statistics to normalize activations, and the backward pass uses the closed-form gradient for parameter updates. We verified that our implementation produces identical outputs to PyTorch's `nn.LayerNorm` under float32 arithmetic.

- **MLP + LN + LSUV**: The combination of both initialization and normalization techniques.

We evaluated all variants on the Adaptive Function Approximation and Single-Shot Qubit Readout benchmarks. The online learning performance metrics are reported in Table 4, and the corresponding FPGA resource utilization and on-chip latency are detailed in Table 5.

First, LSUV provides a modest benefit for vanilla MLPs, confirming that initialization matters. However, the KAN still outperforms the best MLP variant by a wide margin. Furthermore, integrating LayerNorm introduces severe hardware penalties. In fixed-point logic, operations such as square roots and reciprocals required for normalization lead to an excessive utilization of LUTs and DSPs, alongside increased latency. Empirically, we also find that under strict fixed-point constraints, LayerNorm degrades online learning stability and results in reduced performance.

Ultimately, these evaluations demonstrate that KANs, as compared to stronger MLP baselines, still offer superior accuracy at a fraction of the hardware cost while remaining well within the ultra-low latency regime.

*Table 4.* **Online learning performance with extended MLP Baselines.** All variants were evaluated under the same fixed-point precision constraints as the main text.

| Benchmark | Model | Dimension | #Params | Bitwidth $\langle W, I \rangle$ | Metric |
|---|---|---|---|---|---|
| Adaptive Func. Approx. | KAN ($G = 10$) | [1,1] | 13 | $\langle 6, 2 \rangle$ | **13.2** (Regret $\downarrow$) |
| | MLP-P | [1,2,2,1] | 13 | $\langle 6, 2 \rangle$ | 97.6 |
| | MLP-P + LSUV | [1,2,2,1] | 13 | $\langle 6, 2 \rangle$ | 94.9 |
| | MLP-P + LayerNorm | [1,2,2,1] | 21 | $\langle 6, 2 \rangle$ | 112.2 |
| | MLP-P + LN + LSUV | [1,2,2,1] | 21 | $\langle 6, 2 \rangle$ | 127.1 |
| | MLP-L | [1,16,16,1] | 321 | $\langle 6, 2 \rangle$ | 48.3 |
| | MLP-L + LSUV | [1,16,16,1] | 321 | $\langle 6, 2 \rangle$ | 39.5 |
| | MLP-L + LayerNorm | [1,16,16,1] | 385 | $\langle 6, 2 \rangle$ | 145.7 |
| | MLP-L + LN + LSUV | [1,16,16,1] | 385 | $\langle 6, 2 \rangle$ | 81.8 |
| Single-Shot Qubit Readout | KAN ($G = 10$) | [2,7,1] | 273 | $\langle 7, 3 \rangle$ | **92.8%** (Acc $\uparrow$) |
| | MLP-P | [2,20,8,5,1] | 279 | $\langle 10, 3 \rangle$ | 69.8% |
| | MLP-P + LSUV | [2,20,8,5,1] | 279 | $\langle 10, 3 \rangle$ | 70.4% |
| | MLP-P + LayerNorm | [2,20,8,5,1] | 345 | $\langle 10, 3 \rangle$ | 60.2% |
| | MLP-P + LN + LSUV | [2,20,8,5,1] | 345 | $\langle 10, 3 \rangle$ | 51.0% |
| | MLP-L | [2,16,16,16,1] | 609 | $\langle 10, 3 \rangle$ | 62.4% |
| | MLP-L + LSUV | [2,16,16,16,1] | 609 | $\langle 10, 3 \rangle$ | 70.8% |
| | MLP-L + LayerNorm | [2,16,16,16,1] | 705 | $\langle 10, 3 \rangle$ | 61.6% |
| | MLP-L + LN + LSUV | [2,16,16,16,1] | 705 | $\langle 10, 3 \rangle$ | 60.2% |

*Table 5.* **FPGA resource & latency overhead from LayerNorm at 200 MHz clock frequency.** Total Latency includes both the Forward and Backward passes. LSUV initialization is performed offline prior to deployment and therefore introduces no runtime hardware overhead.

| Benchmark | Model | LUTs | FFs | DSPs | Total Latency (ns) |
|---|---|---|---|---|---|
| Adaptive Func. Approx. | KAN | **1,965** | **236** | **3** | **50** |
| | MLP-P | 3,739 | 421 | 8 | 70 |
| | MLP-P + LayerNorm | 10,107 | 2,537 | 16 | 285 |
| | MLP-L | 119,282 | 15,450 | 288 | 215 |
| | MLP-L + LayerNorm | 148,291 | 32,858 | 324 | 530 |
| Single-Shot Qubit Readout | KAN | **45,673** | **6,196** | **63** | **140** |
| | MLP-P | 141,092 | 19,652 | 245 | 225 |
| | MLP-P + LayerNorm | 211,083 | 46,838 | 319 | 1,125 |
| | MLP-L | 180,975 | 92,308 | 880 | 370 |
| | MLP-L + LayerNorm | 428,832 | 83,245 | 663 | 1,620 |

