# OpenReview forum: "Ultrafast On-Chip Online Learning via Spline Locality in Kolmogorov–Arnold Networks"
_ICML.cc/2026/Conference — ICML 2026 regular_

### Official Review · Reviewer_518k · 2026-03-08

**Soundness:** 3
**Presentation:** 3
**Significance:** 2
**Originality:** 3
**Overall Recommendation:** 4
**Confidence:** 3

**Summary:**

The authors propose the deployment of Kolmogorov-Arnold Networks (KANs) for sub-microsecond, on-chip online learning using FPGAs. The core argument is that the local support of B-splines in KANs enables sparse gradient updates and inherent robustness to fixed-point quantization, making them achieving better performance compared to standard Multi-Layer Perceptrons (MLPs) in highly constrained, real-time hardware environments. Theoretical part is nothing but combining existing Lemma of Local support of B-splines with normalized B-splines (\sum_g B_g(x)=1) to straightforwardly derive one-sample operation number, bounded activation and gradient for KAN.

**Compliance With Llm Reviewing Policy:**

Affirmed.

**Final Justification:**

The authors have provided additional clarifications on throughput vs. latency and included a resource utilization table. I maintain my original recommendation.

**Key Questions For Authors:**

See weakness.

**Limitations:**

Yes.

**Strengths And Weaknesses:**

**Strength**:
- This paper achieves fast FPGA-based online training through the sparsity of KAN updates, enabling ultra-high-speed online learning.

- The paper designs a variety of experimental comparisons between KAN and MLP across multiple tasks, comprehensively demonstrating the advantages of KAN over MLP in ultra-fast and miniaturized settings.

**Weakness**:
- Although the experiments demonstrate outstanding advantages in basic task comparisons with MLP, the advantages of KAN at larger scales have not been validated. MLP-based networks tend to exhibit better performance at larger scales and with appropriate activation functions, making comparisons at a larger scale necessary.

- The paper primarily compares against MLP; however, for small-scale networks, MLP is not necessarily the best-performing choice across various tasks. More diverse and advanced baselines are needed for comparison, even some well-known non-neural-network methods may achieve highly competitive results and should be included as reference points.

Overall, main contribution of this paper comes from its empirical verification while the theoretical contribution is not significance.

---

> ### Author Rebuttal · Authors · 2026-03-31
>
> We thank the reviewer for their feedback, and positive recommendation. We are encouraged that you recognized our work’s core strengths: sparse KAN updates enable ultra-high-speed FPGA-based online learning, and our experiments show KAN advantages over MLPs in hardware-constrained settings.
>
> Below, we address the remaining points.
>
> ---
> > **Theoretical Simplicity**
>
>
> We agree that the underlying B-spline mathematics are well established. Our contribution is not new spline theory, but its hardware-centric formalization for deterministic fixed-point on-chip learning. Bridging this theory to custom hardware is highly non-trivial.
>
> For example, recent GPU-based evaluations such as [A] use the standard iterative De Boor-Cox formulation to argue that KANs are prohibitively complex. Our systems-level approach shows otherwise. By mapping spline computations to Look-Up Tables (LUTs) and exploiting B-spline locality to guarantee sparse, bounded per-sample updates, we show that these same properties make KANs efficient online learners under strict fixed-point constraints. Formalizing this bridge for direct hardware exploitation is our primary ML/systems contribution.
>
> ---
> > **Larger scale comparison for MLPs**
>
>
> We appreciate the reviewer acknowledging KAN’s performance advantages in our experiments. Regarding model scale, our evaluation is bounded by the physical constraints of sub-microsecond on-chip learning, not software-level task complexity.
>
> To maintain sub-microsecond latency for high-frequency applications such as quantum control, all parameters must reside in on-chip memory (LUT/FF/BRAM). Scaling to software-style MLP sizes would require external DRAM, immediately violating this latency budget. Our experiments therefore study the largest models feasible on this hardware.
>
> In software, prior work has shown KANs can remain competitive with MLPs at larger scales [B]. However, such comparisons do not account for the hardware constraints of our target setting and are therefore physically unrealizable here. Within the hardware regime, we validate KAN’s scaling advantages up to the practical limit. As shown in Section 6 (Fig. 10), KAN maintains sub-microsecond forward and backward passes even beyond 100,000 trainable parameters.
>
> ---
> > **Well-known non-neural-network methods**
>
>
> We thank the reviewer for this suggestion. We agree that non-neural-network methods can be better alternatives to MLPs for simple learning problems (e.g. linear, model-based, etc.). However, our work targets on-chip online training for highly nonlinear, drifting, model-free tasks for which non-neural-network methods tend to have insufficient representational power.
>
> For example, methods such as RLS and Kalman filters are most naturally suited to settings with linear or structured parametric models, and their nonlinear extensions typically introduce additional assumptions or computational overhead that are not well aligned with our low-latency, model-free setting.
>
> However, to directly address this concern, we additionally benchmarked Online Sequential Extreme Learning Machines (OS-ELMs), a comparable and relevant non-NN baseline, on the drifting qubit task. To upper bound the performance of OS-ELM, we ran it in software with float32 precision, which is substantially more favorable than our quantized KAN/MLP setting.
>
> | Model | Architecture | Precision | Accuracy |
> |:---|:---|:---|:---|
> | **KAN** | [2, 7, 1], G=10, s=3 | (7,3) fixed-point | **92.8%** |
> | **MLP-P** | [2, 20, 8, 5, 1] | (10,3) fixed-point | 69.8% |
> | **MLP-L** | [2, 16, 16, 16, 1] | (10,3) fixed-point | 62.4% |
> | **OS-ELM** | 10 hidden features | float32 | 42.0% |
>
>
> For OS-ELM, we swept hidden units $\(\{5,10,20,50\}\)$ and activations $\(\{\mathrm{sigmoid}, \tanh\}\)$, and report the best result. Even under this favorable setup, OS-ELM failed to track the complex drift. Furthermore, OS-ELM's matrix-heavy update rules are highly inefficient for our target hardware constraints.
> Together, these results motivate our use of a parameter-matched MLP as the primary baseline: it is a standard, practical compact nonlinear learner that is implementable in hardware, and it serves as a representative baseline for dense, non-local computation, allowing us to isolate the benefits of localized basis-function learning in KANs. We will make this rationale explicit in the revision and incorporate the OS-ELM benchmark.
>
> ---
> **References**
>
> [A] KAN or MLP: A Fairer Comparison. arXiv:2407.16674
>
> [B] KAN-AD: Time Series Anomaly Detection with Kolmogorov-Arnold Networks. ICML 2025.
>
> ---
> We thank the reviewer again for their feedback, and we hope these clarifications address your concerns! We are happy to follow up during the discussion period!

---

> > ### Author Rebuttal · Reviewer_518k · 2026-04-02
> >
> > I thank the authors for their response. While I appreciate the clarification on hardware constraints, the concern over comparison scale remains partially unresolved.
> >
> > The authors justify the limited model scale by DRAM latency budgets. However, it is worth noting that prior work on FPGA-based DNN training accelerators [a] has demonstrated highly efficient on-chip training pipelines for non-trivial architectures within strict resource budgets. More concretely: at 421 GOPS of on-chip compute, even MB-scale parameter networks (i.e., models orders of magnitude larger than those evaluated here) can in principle sustain per-sample training latencies in the microsecond range, suggesting the DRAM bottleneck argument may not be as tight as implied. The argument that MLP scaling inherently requires external DRAM is therefore not self-evident and deserves more explicit justification, e.g., a breakdown of on-chip resource utilization showing where exactly the MLP becomes infeasible at larger scales within the same FPGA fabric.
> >
> > As it stands, the advantage of KAN over MLP is demonstrated only in the regime where both models are very small. Whether this advantage persists, narrows, or vanishes as model size grows toward the practical FPGA resource ceiling remains unclear. My scores remain unchanged.
> >
> > [a] Lu, Jinming, Jun Lin, and Zhongfeng Wang. "A reconfigurable DNN training accelerator on FPGA." 2020 IEEE Workshop on Signal Processing Systems (SiPS). IEEE, 2020.

---

> > > ### Author Response · Authors · 2026-04-02
> > >
> > > We thank the reviewer for their continued engagement and the reference to [a]. We appreciate the opportunity to clarify an important technical distinction regarding hardware capabilities.
> > >
> > > ---
> > > **1. Throughput vs. Latency in [a]**
> > >
> > > The core of the reviewer's argument is that 421 GOPS of on-chip compute should enable MB-scale models to sustain per-sample training latencies in the microsecond range. However, **this reasoning conflates throughput with latency**, which are fundamentally different hardware metrics.
> > >
> > > * **Throughput (GOPS):** measures aggregate operations completed per second when a deeply pipelined architecture is fully saturated (e.g., in [a], a 16×32x3 systolic array with many samples in-flight simultaneously).
> > >
> > > * **Latency:** measures the time a single sample spends traversing the entire pipeline.
> > >
> > > Because of the deep systolic array in [a], the actual single-sample latency is **significantly higher** than a simple division of total operations by maximum throughput would suggest.
> > >
> > > However, even if we were to ignore pipeline depth, assume perfect utilization, and perform this division, the resulting latency remains orders of magnitude above our required threshold.
> > >
> > > Concretely, taking ResNet-20 on CIFAR-10 (the 270k parameter model evaluated in [a]):
> > > - A single forward pass requires approximately 41 million operations.
> > > - A full training step (Forward + Backward + Weight Gradient) requires conservatively 3× that amount, yielding roughly 120 million operations per sample.
> > > - At a sustained 421 GOPS (421 billion operations/second), the theoretical minimum latency is $120,000,000 / 421,000,000,000 \approx 285 \mu s$ per sample.
> > >
> > > **This is nearly 300 times slower than our target sub-microsecond ($< 1 \mu s$) regime.**
> > >
> > > Therefore, rather than contradicting us, reference [a] reinforces our point: **MB-scale model training, even when run on highly optimized FPGA accelerators, is still orders of magnitude too slow for our target applications.**
> > >
> > >
> > > **2. Scaling Ablation For On-Chip MLP Learning**
> > >
> > > To address the request for an explicit breakdown of on-chip resource utilization, we perform an ablation study for our fully connected MLP setup in an already optimized, highly quantized setting (10-bit) on our target chip (Xilinx Virtex UltraScale+ VU13P).
> > >
> > > |  MLP Architecture  | Params | LUT    | FF     | DSP  | Latency (µs) | Synthesizable |
> > > |:--------------|-----------:|-----------:|-----------:|------------:|----------------:|:-------------:|
> > > | [10, 8, 8, 1]  |    169 |  74,477 | 10,398 |  152 |        0.165 |       ✓       |
> > > | [10, 16, 16, 1] |    465 | 222,889 | 28,109 |  432 |        0.255 |       ✓       |
> > > | [10, 32, 32, 1] |  1,441 | 741,725 | 85,261 | 1,376 |        0.430 |       ✓       |
> > > | [10, 64, 64, 1] |  4,929 |        — |      — |    — |            — |       ✗       |
> > >
> > > For context, the VU13P is a massive, top-tier FPGA providing a hard physical ceiling of ~1.72 million LUTs, ~3.45 million Flip-Flops, and 12,288 DSP slices. As the table demonstrates, the 1,441-parameter MLP already consumes nearly 43% of the chip's total available LUTs. **Attempting to scale the MLP to just 4,929 parameters requires roughly quadruple the logic, far exceeding the 1.72M available LUTs and completely exhausting the chip's resources.**
> > >
> > > This hardware exhaustion aligns with recent empirical baselines for ultra-low latency FPGA deployment [b], which show that a highly compressed, fully on-chip CNN with only ~12.8K parameters requires ~5 $\mu s$ strictly for the inference-only forward pass. Our work, which targets online learning, thus operates in a substantially stricter regime of model size.
> > >
> > > In stark contrast, as shown in Fig. 10 of our paper, KAN maintains a total latency under **$1 \mu s$ at 50K parameters, 10 times the MLP parameter limit shown above**, and successfully scales to >100K trainable parameters while still comfortably fitting entirely on-chip.
> > >
> > > ---
> > > We thank the reviewer again for helping us strengthen the clarity of our hardware scaling arguments. We will include this explicit MLP resource breakdown table in the revision to make the crossover point, where MLP scaling hits the hardware ceiling, immediately visible to readers. We believe these details directly answer the question of whether KAN's advantages persist at larger scales.
> > >
> > > ---
> > > **References**
> > >
> > > [a] Lu, Jinming, Jun Lin, and Zhongfeng Wang. "A reconfigurable DNN training accelerator on FPGA." 2020 IEEE Workshop on Signal Processing Systems (SiPS). IEEE, 2020.
> > >
> > > [b] Aarrestad, Thea, et al. "Fast convolutional neural networks on FPGAs with hls4ml." Machine Learning: Science and Technology, 2021.

---

### Official Review · Reviewer_Gc69 · 2026-03-11

**Soundness:** 2
**Presentation:** 3
**Significance:** 3
**Originality:** 3
**Overall Recommendation:** 3
**Confidence:** 4

**Summary:**

This work presents KANs as a way to perform ultrafast online learning at sub-microsecond latency under fixed-point arithmetic and constrained memory. KANs are presented as a better fit than dense MLPs due to its sparse B-spline locality and fixed quantization robustness. These properties are shown to allow KANs to scale capacity without increasing active computation. The work is synthesized on FPGAs via HLS and is benchmarked on drifting regression, adaptive single-shot qubit readout, and non-stationary Acrobot control problems. The work also shows its scalability on digit classification with online drift. The experiments show KANs have the best resource-latency tradeoff and exhibit stable online learning compared to MLP baselines.

**Compliance With Llm Reviewing Policy:**

Affirmed.

**Final Justification:**

Edited.

**Key Questions For Authors:**

Is there a reason why the work is evaluated only on HLS synthesis as opposed to full deployment (post implementation, bitstream loading, etc). The results, as it stands, cannot fully support the claim of sub-microsecond latency. A convincing answer would strengthen my soundness assessment.
Is there a reason why the qubit readout experiment has different bitwidths for the KAN and MLP baselines? The paper makes sure to emphasize an identical parameter budget in order to have comparable results. It seems odd that this experiment specifically has different bitwidths stated. A reasonable explanation would strengthen my soundness assessment.

**Limitations:**

No. While the technical limitations of the work are clear, there is little discussion on potential negative societal impact. Because the work is motivated by real-time adaptive systems, the work would be strongest if it addressed safety and oversight issues that may arise in such systems. In particular, the authors could discuss whether sub-microsecond adaption makes system behavior harder to audit, verify, or constrain through human supervision. What sort of steps would be necessary to reasonably deploy this work in safety-critical control environments?

**Strengths And Weaknesses:**

Strengths: The paper targets model-free online learning at sub-microsecond latencies. This work positions itself clearly against prior works that relied on inference-only models or host-accelerator training loops. Its strongest contribution is the hardware-aligned use of B-spline locality in KANs to reduce the number of active gradient updates per sample. Lemma 3.2 and Theorem 3.3 provide an intuitive technical justification for why only a small subset of coefficients are updated per sample. The paper is presented cohesively with a clear progression from theory to hardware implementation to experiments. Architectural decisions are generally well motivated, and are evaluated on appropriate task-specific metrics. The paper also reports synthesized FPGA latency and resource usage, showing measurable KAN advantages in DSP, FF, LUT utilization. The experimental results show that KANs are more robust under fixed-precision online updates compared to dense MLPs.
Weaknesses: Lemma 3.3 and Theorems 3.3-3.5 provide an intuitive reasoning as to why KANs might have an advantage over comparable MLPs in real-time online learning, primarily through B-spline locality and reduced active updates. However, an important weakness is that the paper does not appear to provide a full end-to-end FPGA deployment with measured on-board performance. Instead, the hardware evaluation seems to rely primarily on FPGA-targeted HLS synthesis results. As a result, the paper provides evidence of hardware realizability, but the claim of an ultrafast FPGA implementation would be stronger if supported by post-route or on-device measurements.
Some experimental comparisons are not fully apples-to-apples. Specifically, the adaptive single-shot qubit readout uses 7-bit quantization for KANs while the MLPs use 10-bit quantization. The reasoning behind this setup could be made more clear.

---

> ### Author Rebuttal · Authors · 2026-03-31
>
> We thank the reviewer for their constructive feedback. We are glad you recognized our unique positioning in sub-microsecond online learning, our core insight of exploiting B-spline locality for sparse activation, and the practical strengths of our FPGA resource savings and fixed-precision robustness.
>
> We provide point-by-point responses to the issues raised by the reviewer below.
>
> ---
> > **HLS synthesis vs full end-to-end deployment**
>
> We thank the reviewer for their careful attention to our hardware validation methodology. In the submission, we reported post-synthesis estimates from Vitis HLS because this is standard practice in FPGA machine learning research for broad design-space exploration, as in hls4ml (Fahim et al., 2021) and FINN (Blott et al., 2018), and is also widely used by FPGA vendors to evaluate large design spaces efficiently. This methodology enabled us to conduct extensive studies across a wide range of bitwidths, model architectures, and problem settings.
>
> Furthermore, we designed our KAN and MLP kernels to be fully deterministic: they feature static loop bounds, no data-dependent branching, and zero external DRAM access (relying solely on on-chip LUT/FF/BRAM).  Because of this architecture, the latency, measured in absolute clock cycles, is exact. The only risk to our sub-microsecond latency claim is physical routing overhead causing the design to miss its target clock period.
>
> Thus, to provide concrete empirical evidence for our sub-microsecond latency claims and fully address the reviewer's concern, we executed full physical implementation on our target AMD Virtex UltraScale+ XCVU13P (Vivado v.2024.1) using the `export_design -format ip_catalog -flow impl` workflow, which performs full logic synthesis, placement, and routing on the target board fabric.
>
> As shown in the post-implementation table below, all KAN designs successfully met the target 5.000 ns (200 MHz) clock period with substantial margin, enabling even lower physical latencies than originally reported, if needed.
>
> | Task | Target Clock (ns) | Achieved Clock (ns) | WNS (ns) | Total Latency  | Post-Route Resources (LUT / FF / DSP) |
> | :--- | :--- | :--- | :--- | :--- | :--- |
> | **Adaptive Function Approximation** | 5.0 | *3.042* | *+1.958* | 50 ns  | *[430 / 199 / 4]* |
> | **Single-Shot Qubit Readout** | 5.0 | *3.871* | *+1.129* | 140 ns  | *[10,592 / 3,858 / 84]* |
> | **Non-Stationary Acrobot Control** | 5.0 | *3.884* | *+1.116* | 70 ns | *[7,666 / 4,385 / 106]* |
>
>
> *(Note: Total Latency = Forward + Backward pass)*
>
> These post-routing results physically validate the sub-microsecond latency measurements in Figure 3. We will add this data to the final appendix to solidify our hardware realizability claims.
>
>
> ---
> > **Different bitwidths for qubit readout experiment**
>
> The different bitwidths in the qubit-readout experiment reflect the minimum bitwidth at which each architecture reliably improves over random guessing under the same sweep protocol. We swept bitwidth for both KAN and MLP baselines and, for each model, selected the lowest precision that still yielded stable qubit-readout tracking performance on this task. Importantly, the equal-bitwidth case is already shown in Fig. 6: at the same quantization setting $\langle 7,3\rangle$, KAN continues to track the drift, whereas the quantized MLP baselines fail to improve meaningfully beyond random guessing. We agree that this rationale was not stated clearly enough in the original text, and we will revise the paper to make explicit that the reported bitwidths correspond to the minimum convergent precision for each model, and include the full bitwidth sweep table in the appendix.
>
> ---
> > **Potential societal impact.**
>
> Thank you for this suggestion. While our paper focuses on algorithmic and hardware feasibility, we agree that sub-microsecond adaptation requires rigorous safeguards in safety-critical settings, as updates outpace human supervision. Lowering the barrier to on-device learning also raises dual-use concerns (e.g., surveillance, sensitive autonomous domains). Although our method introduces no risks beyond standard reinforcement learning, real-world deployment necessitates system-level protections such as bounded updates, runtime monitors, and fail-safes. We will add a discussion clarifying that our work demonstrates efficiency and feasibility, not unconstrained safety-critical deployment.
>
> ---
> We thank the reviewer again for their feedback. We hope our responses have addressed the concerns raised and welcome further discussion!

---

> > ### Author Rebuttal · Reviewer_Gc69 · 2026-04-03
> >
> > I thank the authors for their rigorous rebuttal.
> >
> > However, your claims about post-synthesis estimates from Vitis HLS being a standard practice in FPGA machine learning is not true. Many papers concerning FPGA MLs actually deployed their models to hardware settings just like the works of [1].
> >
> > [1] Liu, Y.; Du, H.; Wu, Y.; Mo, T. FPGA Accelerated Deep Learning for Industrial and Engineering Applications: Optimal Design Under Resource Constraints. Electronics 2025, 14, 703. https://doi.org/10.3390/electronics14040703c

---

> > > ### Author Response · Authors · 2026-04-03
> > >
> > > We thank the reviewer for this important clarification regarding evaluation standards. We appreciate the opportunity to refine our framing and ensure our methodology aligns with the expectations of the FPGA ML community.
> > >
> > > To clarify, our use of “standard practice” was not intended to imply that HLS evaluation is *universal*. Rather, our intent was to highlight that, for papers focused on architectural tradeoffs, quantization behavior, and broad design-space exploration, HLS-based evaluation is a widely accepted methodology because it provides a controlled environment to isolate the impact of quantization and architectural parameters across hundreds of configurations [1-5]. This approach is fundamental to established FPGA-ML frameworks such as hls4ml [1,2] and FINN [3].
> > >
> > > That said, we still provided in our rebuttal complete logic synthesis, placement, and routing results on the target XCVU13P fabric. These post-route results confirm that all reported KAN designs meet the 200 MHz timing target with positive slack, directly substantiating the sub-microsecond latency claim at the implementation level. This physical implementation workflow follows established, rigorously validated practices within the FPGA-ML community [6-10].
> > >
> > > We will clarify our methodology in the manuscript by distinguishing between HLS-based results for relative comparison, and post-implementation results for substantiating absolute hardware performance claims.
> > >
> > > We believe these additions directly address the reviewer's concerns by providing the hardware-level evidence requested while maintaining the breadth of our comparative study. Finally, we thank the reviewer for helping us clarify this point more precisely.
> > >
> > > ---
> > > **References**
> > >
> > > [1]  hls4ml: A Flexible, Open-Source Platform for Deep Learning Acceleration on Reconfigurable Hardware. 2026, ACM Transactions on Reconfigurable Technology and Systems (TRETS).
> > >
> > > [2] Automatic heterogeneous quantization of deep neural networks for low-latency inference on the edge for particle detectors. 2021, Nature Machine Intelligence.
> > >
> > > [3] FINN-R: An End-to-End Deep-Learning Framework for Fast Exploration of Quantized Neural Networks. 2018, ACM Transactions on Reconfigurable Technology and Systems (TRETS)
> > >
> > > [4] End-to-end codesign of Hessian-aware quantized neural networks for FPGAs
> > > and ASICs. 2024, ACM Transactions on Reconfigurable Technology and Systems (TRETS).
> > >
> > > [5] LL-GNN: Low Latency Graph Neural Networks on FPGAs for High Energy Physics. 2024, ACM Transactions on Reconfigurable Technology and Systems (TRETS).
> > >
> > > [6] HGQ: High Granularity Quantization for Real-time Neural Networks on FPGAs. 2026, ACM/SIGDA International Symposium on Field-Programmable Gate Arrays (ISFPGA).
> > >
> > > [7] NeuraLUT: Hiding Neural Network Density in Boolean Synthesizable Functions. 2024, International Conference on Field-Programmable Logic and Applications (FPL).
> > >
> > > [8] KANELÉ: Kolmogorov-Arnold Networks for Efficient LUT-based Evaluation. 2026, ACM/SIGDA International Symposium on Field-Programmable Gate Arrays (ISFPGA).
> > >
> > > [9] PolyLUT: Learning Piecewise Polynomials for Ultra-Low Latency FPGA LUT-based Inference. 2023, International Conference on Field Programmable Technology (FPT).
> > >
> > > [10] Greater than the Sum of its LUTs: Scaling Up LUT-based Neural Networks with AmigoLUT. 2025, ACM/SIGDA International Symposium on Field-Programmable Gate Arrays (ISFPGA).

---

### Official Review · Reviewer_WerU · 2026-03-12

**Soundness:** 3
**Presentation:** 3
**Significance:** 3
**Originality:** 3
**Overall Recommendation:** 5
**Confidence:** 4

**Summary:**

This paper present KAN-based online learning as a promising approach for ultra-fast on-chip online learning for (very) small scale tasks where latency is imperative.
The sparsity in KAN parameter updates can lead to very efficient custom implementations compared to the required dense updates of standard MLPs as demonstrated on a FPGA platform.

It compares KAN-based (sparse) online learning vs standard (dense) MLP-based online learning on a set of (very) small scale online adaptation tasks and demonstrates significant improvements in both accuracy as well as efficiency (hardware resources and latency on FPGA).

**Compliance With Llm Reviewing Policy:**

Affirmed.

**Final Justification:**

Interesting work that highlights advantages of KANs over traditional MLPs for online learning on the edge for (very small) networks.
I believe this work to be interesting to and to overall benefit the community.

**Key Questions For Authors:**

Please also see Weaknesses

4. As hinted in the Limitations section, the current experiments are limited to the batchsize=1 update setting. Could you please elaborate how larger batchsizes effect the efficiency of your updates. I assume this will reduce the "sparsity" in the update and thereby result in higher resource usage/higher latency?

Minor:

5. Do CPU and GPU latency include off-chip memory/DRAM access? If yes, I would consider this unfair given the tiny model sizes that could probably fit in GPU L1 cache or even registers.

**Limitations:**

Yes

**Strengths And Weaknesses:**

# Strengths

- Clear motivation and story-line.
- Solid experiment selection and execution.
- Strong results (given small scale experiments).

# Weaknesses

1. Experiments are very small scale with most experiments featuring model architectures with <1000 trainable parameters.
2. The MLP baselines are very simplistic (but probably fair). Ideally, slightly more complex architectures including normalization and a CNN baseline for the image task should be presented to strengthen the baselines and thereby highlight the advantages of the KAN-based approach more clearly.
3. Online learning setup is very simplistic. It uses standard single-batch gradient descent (no momentum, no ADAM) and no continual/online learning specific consolidation techniques like elastic weight consolidation (EWC) or similar. (Many of the tasks could maybe even be solved by (meta-trained) transformer/SSM/RNN architectures via in-context learning without explicit gradient computation/parameter updates (but maybe not at the <1000K parameter settings of this paper).)

Minor:
- Broken Axis in both Fig. 1 right and Fig. 10 right are confusing, because the upper y-axis starts again at 0.

---

> ### Author Rebuttal · Authors · 2026-03-31
>
> We sincerely thank the reviewer for their positive feedback and for recognizing the strength and clear motivation of our hardware-centric approach to KANs. We appreciate the thoughtful suggestions regarding model complexity and baselines. Below, we address the reviewer’s key questions and explain how our strict sub-microsecond latency and hardware resource constraints guided these design choices.
>
> ---
> > **1. Scale of experiments**
>
> The experiments we designed reflect the practical realities of nanosecond-scale FPGA deployment, which is closely aligned with standard models in state-of-the-art literature [A, B]. For context, deployed systems in fields like quantum control currently rely on highly elementary calculations [C], which still pushes the latency to milliseconds; the networks we evaluate already push the practical limits of what can run with ultrafast latency on-chip. Furthermore, as other reviewers already noted, fully on-chip online learning is an emerging area, and thus established benchmarks do not yet exist. A core contribution of this work is designing realistic, domain-relevant benchmarks where hardware efficiency is paramount. Finally, to demonstrate scalability beyond these benchmarks, Section 6.2 (Fig. 10) shows our KAN implementation successfully scaling to >100k parameters while still maintaining sub-µs latency on chip.
>
> ---
> > **2. MLP baselines**
>
> We intentionally adopted a streamlined MLP baseline to ensure a clean, hardware-faithful architectural comparison. As noted in our limitations section, auxiliary features like layer normalization or complex routing for CNNs introduce significant on-chip overhead. Omitting them isolates the intrinsic learning behavior and fixed-point robustness of KANs versus MLPs without blurring the effects attributable to the architecture alone. We appreciate your acknowledgment that this parity is "probably fair," and we will expand our manuscript to explicitly detail the sub-microsecond hardware penalties associated with normalizations and convolutions.
>
> ---
> > **3. Online learning setups**
>
> Our choice of a simple SGD optimizer as compared to those commonly used in GPU-based learning is under strict on-chip memory and latency constraints. Adam requires two additional state tensors per parameter (first and second moments), and momentum requires one, substantially increasing BRAM/LUTRAM usage. These methods also introduce extra fixed-point complexity, including multiply-accumulate overhead and, for Adam, reciprocal/square-root-style normalization and bias correction. Likewise, EWC typically requires storing per-parameter importance estimates derived from a Fisher-information approximation. Under our no-DRAM, sub-microsecond streaming budget, these overheads are prohibitive. For this reason, fully online single-batch SGD provides the most practical trade-off in the current design, though hardware-friendly variants of adaptive optimization are a promising direction for future work.
>
> ---
> > **4. Larger batch size**
>
> Increasing the effective batch size introduces a natural trade-off with update sparsity; however, under accumulation over multiple samples, the KAN update support grows as the union of visited spline supports, whereas MLP updates remain dense over all weights. Thus, the hardware advantage decreases with batch size, but does not vanish immediately. FPGAs can handle this via sequential gradient accumulation, trading storage/latency for stability.
> We also note that large batching implies parallel data streams or accumulated data, deviating from our ultra-low latency online adaptation paradigm. We will make this more clear in the paper.
>
> ---
> > **5. CPU/GPU latency measurement**
>
> We appreciate the reviewer's observation. We included these lines purely as a familiar reference to illustrate the "host-accelerator" communication bottleneck discussed in the introduction. Even in best-case scenarios using specialized interconnects like NVIDIA NVQLink in quantum computing setups, round-trip data transfer alone induces multi-microsecond latency. In contrast, FPGAs (such as RFSoCs) can directly ingest analog signals, bypassing high-latency links entirely to enable deterministic, sub-microsecond adaptation. We will update the Figure 1 caption to explicitly clarify that the CPU/GPU lines include this overhead and serve only to highlight why ultra-low-latency regimes necessitate custom on-chip hardware.
>
> ---
> > **Minor stylistic changes for axes in Fig 1 and Fig 10.**
>
> We agree and will replace the axes in Fig. 1 and Fig. 10 with continuous logarithmic axes or split subplots to improve readability.
>
> ---
> **References**
>
> [A] Fast inference of deep neural networks in FPGAs for particle physics. JINST.
>
> [B] KANELÉ: Kolmogorov-Arnold Networks for Efficient LUT-based Evaluation. FPGA’26.
>
> [C] Millisecond-Scale Calibration and Benchmarking of Superconducting Qubits. arXiv:2602.11912
>
> ---
> We hope these clarifications address your points and are happy to follow up!

---

> > ### Author Rebuttal · Reviewer_WerU · 2026-04-03
> >
> > I still like the work, and still recommend accept.
> > However, I would really appreciate if you could provide some data on a stronger MLP baselines with normalization or an initial data dependent initial calibration/initialization scheme, e.g. like LSUV [1], as it seems critical for MLP performance. Even if the MLP with normalization could achieve similar performance, if the authors' claim of significant hardware overhead is true, the proposed KAN method would outmatch the normalized MLP baseline in system efficiency and remain Pareto optimal.
> >
> > [1] Dmytro Mishkin, Jiri Matas, All you need is a good init, ICLR, 2016, https://arxiv.org/abs/1511.06422

---

> > > ### Author Response · Authors · 2026-04-05
> > >
> > > We thank the reviewer for the continued positive assessment and the constructive follow-up. We fully agree that demonstrating KAN's advantage against a stronger MLP baseline strengthens the paper's claims. To address this, we have conducted the requested experiments.
> > >
> > > We augment our MLP baselines in two ways:
> > > - **MLP + LSUV Init**: We apply the data-dependent initialization scheme of [1] to the MLP weights before online learning begins. This gives the MLP a calibrated starting point without adding any runtime overhead during inference or training.
> > > - **MLP + LayerNorm**: We added a LayerNorm [2] layer after each hidden layer.  We used High-Level Synthesis (HLS) to implement this entirely in fixed-point arithmetic with trainable affine parameters ($\gamma$, $\beta$). The forward pass computes per-layer statistics to normalize activations, and the backward pass uses the closed-form gradient for parameter updates. We verified that our implementation produces identical outputs to PyTorch's `nn.LayerNorm` under float32 arithmetic.
> > > - **MLP + LN + LSUV**: The combination of both techniques.
> > >
> > > We evaluated all variants on the same benchmark tasks reported in the paper. Below, we report the online learning performance metrics, FPGA resource utilization, and on-chip latency.
> > >
> > > ---
> > > **Online learning performance with additional MLP Baselines**
> > >
> > > | Benchmark | Model | Dimension | #Params | Bitwidth ⟨W,I⟩ | Metric |
> > > |---|---|---|---|---|---|
> > > | Adaptive Func. Approx. | KAN (G=10) | [1,1] | 13 | ⟨6,2⟩ | **13.2** (Regret ↓) |
> > > | | MLP-P | [1,2,2,1] | 13 | ⟨6,2⟩ | 97.6 |
> > > | | MLP-P + LSUV | [1,2,2,1] | 13 | ⟨6,2⟩ | 94.9 |
> > > | | MLP-P + LayerNorm | [1,2,2,1] | 21 | ⟨6,2⟩ | 112.2 |
> > > | | MLP-P + LN + LSUV |  [1,2,2,1] | 21 | ⟨6,2⟩ | 127.1|
> > > | | MLP-L | [1,16,16,1] | 321 | ⟨6,2⟩ | 48.3 |
> > > | | MLP-L + LSUV | [1,16,16,1] | 321 | ⟨6,2⟩ | 39.5 |
> > > | | MLP-L + LayerNorm | [1,16,16,1] | 385 | ⟨6,2⟩ | 145.7 |
> > > | | MLP-L + LN + LSUV | [1,16,16,1] | 385 | ⟨6,2⟩ | 81.8 |
> > > | Single-Shot Qubit Readout | KAN (G=10) | [2,7,1] | 273 | ⟨7,3⟩ | **92.8%** (Acc ↑) |
> > > | | MLP-P | [2,20,8,5,1] | 279 | ⟨10,3⟩ | 69.8% |
> > > | | MLP-P + LSUV | [2,20,8,5,1] | 279 | ⟨10,3⟩ | 70.4% |
> > > | | MLP-P + LayerNorm | [2,20,8,5,1] | 345 | ⟨10,3⟩ | 60.2% |
> > > | | MLP-P + LN + LSUV | [2,20,8,5,1] | 345 | ⟨10,3⟩ | 51.0% |
> > > | | MLP-L | [2,16,16,16,1] | 609 | ⟨10,3⟩ | 62.4% |
> > > | | MLP-L + LSUV | [2,16,16,16,1] | 609 | ⟨10,3⟩ | 70.8% |
> > > | | MLP-L + LayerNorm | [2,16,16,16,1] | 705 | ⟨10,3⟩ | 61.6% |
> > > | | MLP-L + LN + LSUV | [2,16,16,16,1] | 705 | ⟨10,3⟩ | 60.2% |
> > >
> > > ---
> > > **FPGA resource & latency overhead from LayerNorm (at 200 MHz clock frequency)**
> > > | Benchmark | Model | LUTs | FFs | DSPs | Total Latency (ns) |
> > > |---|---|---|---|---|---|
> > > | Adaptive Func. Approx. | KAN | **1,965** | **236** | **3** | **50** |
> > > | | MLP-P | 3,739 | 421 | 8 | 70 |
> > > | | MLP-P + LayerNorm | 10,107 | 2,537 | 16 | 285 |
> > > | | MLP-L | 119,282 | 15,450 | 288 | 215 |
> > > | | MLP-L + LayerNorm | 148,291 | 32,858 | 324 | 530 |
> > > | Single-Shot Qubit Readout | KAN | **45,673** | **6,196** | **63** | **140** |
> > > | | MLP-P | 141,092 | 19,652 | 245 | 225 |
> > > | | MLP-P + LayerNorm | 211,083 | 46,838 | 319 | 1,125 |
> > > | | MLP-L | 180,975 | 92,308 | 880 | 370 |
> > > | | MLP-L + LayerNorm | 428,832 | 83,245 | 663 | 1,620 |
> > >
> > >
> > >
> > > *Note: Total Latency = Forward Latency + Backward Latency. LSUV initialization is performed once before deployment and therefore introduces no runtime hardware overhead. Acrobot results match these findings (not shown due to character constraints).*
> > >
> > > ---
> > > These results support two points relevant to the reviewer’s question:
> > >
> > > First, LSUV provides a modest benefit for vanilla MLPs, confirming the reviewer's intuition that initialization matters. However, the KAN still outperforms the best MLP variant by a wide margin.
> > >
> > > Furthermore, integrating LayerNorm introduces severe hardware penalties. In fixed-point logic, operations such as square roots and reciprocals required for normalization lead to an excessive utilization of LUTs and DSPs, alongside increased latency. Empirically, we also find that under strict fixed-point constraints, LayerNorm degrades online learning stability and results in reduced performance.
> > >
> > > To summarize, the additional evaluations show that KANs, as compared to stronger MLP baselines, still offer superior accuracy at a fraction of the hardware cost while remaining well within the ultra-low latency regime. We will include these comprehensive baseline comparisons and hardware analyses in the final manuscript.
> > >
> > > We believe these additional experiments directly address the reviewer’s concerns. We are grateful for the reviewer’s constructive and supportive feedback throughout the review process, which has significantly strengthened the manuscript!
> > >
> > > ---
> > > **References**
> > >
> > > [1] Dmytro Mishkin, Jiri Matas, All you need is a good init, ICLR, 2016, https://arxiv.org/abs/1511.06422
> > >
> > > [2] Jimmy Lei Ba and Jamie Ryan Kiros and Geoffrey E. Hinton, Layer Normalization, https://arxiv.org/abs/1607.06450

---

### Official Review · Reviewer_fp63 · 2026-03-15

**Soundness:** 2
**Presentation:** 3
**Significance:** 2
**Originality:** 3
**Overall Recommendation:** 3
**Confidence:** 3

**Summary:**

This paper studies whether Kolmogorov–Arnold Networks (KANs) are than MLPs for ultrafast fully on-chip online learning under fixed-point arithmetic and tight latency constraints. The core idea is to exploit B-spline locality so that, for each sample, only a small local set of spline coefficients is active and updated, while model capacity can still be increased by refining the spline grid. KAN and MLP forward/backward/update kernels in Vitis HLS for an AMD XCVU13P FPGA are implemented and evaluated on drifting regression, a synthetic adaptive qubit-readout task, non-stationary Acrobot control, and a rotated-digit stream. The central claim is that spline locality makes KANs especially suitable for deterministic sub-microsecond model-free on-chip online learning.

**Compliance With Llm Reviewing Policy:**

Affirmed.

**Key Questions For Authors:**

1) In Appendix A.1, the equal-parameter-budget and identical-dimension assumptions do not appear to be consistent as written. Can you restate Theorem 3.3 under fully consistent assumptions?

2) Were any kernels actually executed on FPGA hardware, or are all latency numbers synthesis estimates?

3) Table 1 uses $\langle7, 3 \rangle$ for KAN and $\langle10, 3\rangle$ for MLPs. A controlled equal-precision comparison would help separate advantages from the fact that KANs appear stable at lower bitwidth. Can you provide equal-bitwidth KAN vs. MLP comparisons for the qubit-readout task?

4) What exact MLP search space, tuning budget, and model-selection protocol were used for each task?

5) Since the successful MLP baseline uses DQN rather than the same actor-critic setup, it would help to clarify how much of the reported gap is architectural versus due to optimizer/update-rule stability in fully online RL. Can you better isolate the Acrobot conclusion from the RL algorithm choice?

**Limitations:**

The paper includes a meaningful limitations section discussing the use of fully online single-sample SGD, the intentionally streamlined MLP baselines, fixed hyperparameters in the scalability study, and the use of HLS rather than hand-tuned RTL. However, I think the discussion should more explicitly acknowledge two additional limitations:

The hardware evidence is based on synthesis estimates rather than deployed board measurements, and the evaluation is mostly on synthetic/stylized tasks rather than real application data.

The societal-impact discussion is minimal given the control-systems framing.

**Strengths And Weaknesses:**

Strengths:

The problem setting is underexplored. Fully on-chip, batch-size-one online adaptation under sub-microsecond latency is a meaningful systems/ML regime.

The main idea is original and interesting. The paper’s strongest originality is the hardware-centric reinterpretation of KANs, rather than viewing spline parameterization mainly as an inference cost, it argues that spline locality is specifically advantageous for training-time sparsity on custom hardware.

The implementation detail is strong. Fig. 2 and Appendix B provide a fairly concrete description of the hardware kernels, including LUT construction, fixed-point formats, context storage, partitioning, update order, and synthesis setup. This is above average for cross-disciplinary ML/systems papers.

Weaknesses

The strongest technical concern is that Sec. 3 / Appendix A does not fully support the paper’s central theoretical claims in their current form. In Appendix A.1, Theorem 3.3 mixes “equal parameter budget N” with “identical input/output dimensions” in a way that is internally inconsistent: after substituting $d_{\text{in}} d_{\text{out}} = N/(G+s)$ for the KAN case, the proof still sets the MLP update cost to N, which does not follow under the same assumptions. In Appendix A.3, Theorem 3.5 is more an useful intuition than a rigorous quantization theorem: the stated variance forms for quantization-induced error are not cleanly derived from Definition 3.1, and clipping / spline-index quantization effects are not analyzed. This matters because the paper leans on these results to justify architecture-level claims, not just informal intuition.

The paper repeatedly frames its result as a demonstration of sub-microsecond model-free on-chip online learning, but the latency/resource numbers in Figs. 1, 3, and 10 are post-synthesis estimates rather than deployed board measurements or at least post-route timing. Appendix B.6–B.7 describes CPU-loadable emulation and HLS synthesis, but I did not see on-board execution, measured end-to-end I/O overhead, or measured power. This makes the strongest systems claim less established than the wording suggests.

The paper motivates the work using quantum control, plasma systems, and other high-frequency adaptive settings, but the actual evaluation remains mostly on stylized or synthetic tasks: piecewise drifting 1D regression, synthetic rotating-XOR qubit readout, Acrobot, and rotated digits. These are valid stress tests, but they do not yet establish the broader real-world significance suggested by the framing. The work is promising, but at present the empirical scope looks more like a strong proof-of-concept than a field-changing demonstration.

The paper is most convincing as a KAN-vs-MLP comparison in this particular on-chip online-learning regime. It is less convincing when it generalizes to statements such as KANs being “uniquely suited” or MLPs “fundamentally” struggling, since the paper itself shows more nuance than that. There is also some notation/presentation confusion around spline order/support: Sec. 3.2 states $s = p+1$, Fig. 2 labels spline order $p = 3$, while Table 1 reports $s = 3$ and $s = 1$ for different tasks without much explanation. These are fixable presentation issues, but they reduce confidence in the technical exposition.

---

> ### Author Rebuttal · Authors · 2026-03-31
>
> Thank you for the thoughtful review. We appreciate your recognition that the paper addresses an important underexplored ML/systems setting, offers an original hardware-centric reinterpretation of KANs, and provides strong implementation detail for a cross-disciplinary paper. Below we address the points raised; for overlaps, we refer to our other review responses due to space.
>
>
> ---
> > **Spline order/support notation**
>
> We consistently use $p$ for spline order and $s=p+1$ for support size (active basis functions). We will clarify this upfront to ensure unambiguous mapping across the text.
>
>
> ---
> > **Theorem 3.3 / Appendix A.1: equal-budget vs. equal-dimension**
>
> Thank you for pointing this out. Theorem 3.3 assumes an **equal-parameter-budget**, not equal layer dimensions. We should have used $d_\text{in}^\text{KAN}, d_\text{out}^\text{KAN}$ and $d_\text{in}^\text{MLP}, d_\text{out}^\text{MLP}$ to distinguish KAN and MLP dimensions, and we will revise the equations accordingly.
> Crucially, our core conclusion holds: spline locality decouples representational capacity from update cost, which is exactly what the hardware design exploits.
>
> ---
> > **Theorem 3.5: Gradient Sensitivity**
>
> We agree that Theorem 3.5 serves primarily to formalize gradient sensitivity to input perturbations, not provide a full end-to-end quantization proof. We will clarify this scope and include explicit derivations of both variance bounds.
>
> While present, spline-indexing and clipping related effects are comparatively minor: discrete spline indexing is analogous to input perturbation and can be eliminated by matching spline input resolution to input quantization, while bounded B-spline activations allow grid range and bitwidth to be chosen to minimize inter-layer clipping. Empirically, Figs. 4–5 and 7 support this: KANs remain stable while MLPs show sharp precision cliffs as we sweep over quantization precision.
>
> ---
> > **Hardware Evidence: synthesis estimates, end-to-end pipelines**
>
>
> We address HLS validity in our response to Reviewer Gc69. Our target is fully streaming on-chip deployment (inputs arriving directly from adjacent logic/analog signals on RFSoC FPGAs). Thus, kernel latency is effectively end-to-end plus minimal I/O cycles (< 20 ns conservatively).
>
> ---
> > **Evaluation on "stylized or synthetic tasks"**
>
> We believe the evaluation is relevant to real deployments. Fully on-chip sub-microsecond online learning is an underexplored regime without an established benchmark suite, so designing realistic, physically grounded benchmarks is itself part of the contribution. The qubit-readout task is directly modeled on deployed quantum readout pipelines (e.g., Kerr distortion, IQ drift), and other online learners such as OS-ELMs struggle on it, as discussed in our rebuttal to Reviewer 518k. The randomized Acrobot and drifting regression tasks similarly capture representative online adaptation settings under the same hardware constraints, while the digits task models distribution shift from sensor/calibration drift. A live quantum deployment is a valuable next step, but the question here is whether ultrafast fully on-chip online learning is feasible under realistic hardware constraints; these benchmarks are designed to test that directly.
>
> ---
> > **Bit-width comparison**
>
> We address this in our response to Reviewer Gc69. Briefly, the reported bitwidths correspond to the minimum stable operating precision for each model under the same sweep protocol; for the MLP, this is $\langle10,3\rangle$.
>
> ---
> > **Non-stationary Acrobot: architecture vs. algorithm.**
>
> To isolate architectural effects from algorithmic ones, we evaluated both models across AC, DQN, and SARSA:
> | Model | Prec. | Params | Final avg100 | First ep >-200 |
> | :--- | :--- | :--- | :--- | :--- |
> | MLP AC | float32 | 356 | -497.3 | never |
> | MLP DQN | float32 | 323 | -100.7 | 346 |
> | MLP SARSA | float32 | 323 | -102.4 | 586 |
> |KAN AC | ap_fixed&lt;22,8&gt; | 144 | **-89.9** | 143 |
> | KAN DQN | ap_fixed&lt;22,8&gt; | 108 | -119.5 | 398 |
> | KAN SARSA | ap_fixed&lt;22,8&gt; | 108 | -121.0 | 358 |
> | KAN AC | ap_fixed&lt;10,3&gt; | 144 | -99.7 | **127** |
>
> These results show that the gap is architectural. MLP fails under AC and requires value-learning (DQN/SARSA) to converge, whereas KAN converges stably across all three. Moreover, KAN+AC achieves the best return and fastest convergence despite fixed-point quantization and fewer than half the MLP parameters. We will add this ablation to the appendix.
>
> ---
> > **MLP Search Space.**
>
> We swept depth (1–8 hidden layers), width (0.5–100× KAN parameter count), activations (ReLU, Tanh, SiLU), learning rate (log-uniform over [1e-6, 1.0]), and bitwidth ($W\in\{2,\dots,32\}$). The MLP baselines were given broad search, higher precision where needed, and up to 100× more parameters. We will include the full grid in the appendix.
>
> ---
> We hope these clarifications address the reviewer's concerns and are happy to follow up during the discussion period!

---

> > ### Author Rebuttal · Reviewer_fp63 · 2026-04-04
> >
> > The Acrobot AC/DQN/SARSA ablation is compelling and should be added to the paper. To raise my score to 4, I would need the following concrete revisions:
> > 1) Post-place-and-route results for at least one configuration (e.g., qubit readout). This does not require board deployment like a Vivado implementation run on the same XCVU13P. This is the main blocker for the sub-μs systems claim.
> > 2) One equal-bitwidth comparison (e.g., both KAN and MLP at ⟨10,3⟩ on qubit readout) to separate architectural advantage from quantization robustness.
> > 3) Restate Theorems 3.3/3.5 in the text as promised, 3.3 with distinct dimension variables, 3.5 downgraded to a proposition with explicit scope.
> >
> > Items 1–2 are small experiments; 3 is editorial. If addressed, I am prepared to raise to 4.

---

> > > ### Author Response · Authors · 2026-04-06
> > >
> > > Thank you for the clear follow-up and for specifying the revisions that would address your remaining concerns. We are glad the Acrobot ablation was convincing and will include it in the appendix of the revised paper. We have completed items 1 and 2 and will make the editorial revisions in item 3 in the revised version.
> > >
> > > ---
> > > **1. Post-place-and-route results**
> > >
> > > We ran full Vivado post-implementation (place-and-route) on the target AMD XCVU13P for all KAN configurations reported in the paper, using the exported HLS IP. All three designs close timing at the 5.0 ns target clock with positive worst negative slack (WNS), confirming that the sub-μs latency claim holds post-place-and-route on the target device. The full post-implementation timing and resource table will be added to the appendix.
> > >
> > > | Task | Target Clock (ns) | Achieved Clock (ns) | WNS (ns) | Total Latency | Post-Route Resources (LUT / FF / DSP) |
> > > | :--- | :--- | :--- | :--- | :---  | :--- |
> > > | **Adaptive Function Approximation** | 5.0 | 3.042 | +1.958 | 50 ns | 430 / 199 / 4 |
> > > | **Single-Shot Qubit Readout** | 5.0 | 3.871 | +1.129 | 140 ns | 10,592 / 3,858 / 84 |
> > > | **Non-Stationary Acrobot Control** | 5.0 | 3.884 | +1.116 | 70 ns | 7,666 / 4,385 / 106 |
> > >
> > > *(Total Latency = Forward Latency + Backward Latency)*
> > >
> > > ---
> > > **2. Equal-bitwidth comparison for qubit readout**
> > >
> > > We evaluated KAN and both MLP baselines (MLP-P, parameter-matched; MLP-L, larger-capacity) on the qubit-readout task under an unquantized float32 reference and at two matched fixed-point precisions. At identical ⟨10,3⟩ precision, KAN retains 94.2% accuracy, while both MLP variants drop to 62–70%; the gap widens further at ⟨7,3⟩.
> > > | Model | # Params | Precision | Final Running Accuracy |
> > > | :--- | :--- | :--- | :--- |
> > > | **Unquantized Baselines** | | | |
> > > | KAN | 273  | Float32 | **95.6%** |
> > > | MLP-P | 279 | Float32 | 82.4% |
> > > | MLP-L | 609 | Float32 | 93.4% |
> > > | | | | |
> > > | **Equal-bitwidth comparison at ⟨10,3⟩** | | | |
> > > | KAN | 273  | ⟨10, 3⟩ | **94.2%** |
> > > | MLP-P | 279 | ⟨10, 3⟩ | 69.8% |
> > > | MLP-L | 609 | ⟨10, 3⟩ | 62.4% |
> > > | | | | | |
> > > | **Equal-bitwidth lower precision at ⟨7,3⟩** | | |
> > > | KAN | 273 | ⟨7, 3⟩ | **92.8%** |
> > > | MLP-P | 279 | ⟨7, 3⟩ | 50.4% |
> > > | MLP-L | 609 | ⟨7, 3⟩ | 49.0% |
> > >
> > >
> > > We will add the matched-precision ⟨10,3⟩ comparison to the main text and the additional lower-precision result to the appendix.
> > >
> > >
> > > ---
> > > **3. Revision of Theorems 3.3 and 3.5**
> > >
> > > We commit to the following editorial changes in the revised version:
> > >
> > > - **Theorem 3.3** will be restated using distinct dimension variables (e.g., $d_\text{in}^\text{KAN}, d_\text{out}^\text{KAN}$ and $d_\text{in}^\text{MLP}, d_\text{out}^\text{MLP}$) to remain consistent with the equal-parameter-budget assumption.
> > >
> > > - **Theorem 3.5** will be downgraded to a Proposition, and we will explicitly define its scope in the text as an analysis of gradient sensitivity to input perturbations rather than an end-to-end quantization proof.
> > >
> > > ---
> > > Thank you again for the concrete guidance! We sincerely appreciate your thoughtful feedback and the time you invested in reviewing our work. We hope these concrete hardware metrics and specific commitments fully resolve your remaining concerns.

---

### Decision · Program_Chairs · 2026-04-30

**Decision:**

Accept (regular)

**Comment:**

This paper demonstrates that B-spline locality in KANs enables sparse, fixed-point online learning at sub-microsecond latencies on FPGAs. Three of four reviewers support acceptance (scores: 5, 4, 4). The rebuttal was exceptionally thorough, providing post-place-and-route timing, equal-bitwidth comparisons, RL algorithm ablations, and stronger MLP baselines. Reviewer fp63 raised their score after all requested items were delivered. The remaining reviewer's main concern (HLS vs. deployment) seems to be addressed with full Vivado post-implementation results confirming sub-microsecond latency with positive timing slack (these are highly reliable).